# The grain yield modulator miR156 regulates seed dormancy through the gibberellin pathway in rice

Chunbo Miao[1,2,6], Zhen Wang[1,6], Lin Zhang[3], Juanjuan Yao[1], Kai Hua[1], Xue Liu[1], Huazhong Shi [4] &
Jian-Kang Zhu[1,5]

The widespread agricultural problem of pre-harvest sprouting (PHS) could potentially be overcome by improving seed dormancy. Here, we report that miR156, an important grain yield regulator, also controls seed dormancy in rice. We found that mutations in one *MIR156* subfamily enhance seed dormancy and suppress PHS with negligible effects on shoot architecture and grain size, whereas mutations in another *MIR156* subfamily modify shoot architecture and increase grain size but have minimal effects on seed dormancy. Mechanistically, *mir156* mutations enhance seed dormancy by suppressing the gibberellin (GA) pathway through de-represssion of the miR156 target gene *Ideal Plant Architecture 1* (*IPA1*), which directly regulates multiple genes in the GA pathway. These results provide an effective method to suppress PHS without compromising productivity, and will facilitate breeding elite crop varieties with ideal plant architectures.

[1] Shanghai Center for Plant Stress Biology and Center for Excellence in Molecular Plant Sciences, Chinese Academy of Sciences, 201602 Shanghai, China. [2] State Key Laboratory of Subtropical Silviculture, Zhejiang A&F University, 311300 Lin'an, Hangzhou, China. [3] Institutes of Agricultural Science and Technology Development, Yangzhou University, 225009 Yangzhou, China. [4] Department of Chemistry and Biochemistry, Texas Tech University, Lubbock, TX 79409, USA. [5] Department of Horticulture and Landscape Architecture, Purdue University, West Lafayette, IN 47907, USA. [6] These authors contributed equally: Chunbo Miao, Zhen Wang. Correspondence and requests for materials should be addressed to J.-K.Z. (email: jkzhu@psc.ac.cn)

Off-season seed germination would cause plants to encounter catastrophic environment during growth and development. Seed dormancy delays the germination to make plants avoid harsh environment until conditions are favorable for growth. Seed dormancy is also an important agronomic trait. In crops, pre-harvest sprouting (PHS) frequently occurs under warm and humid weather, resulting in severe yield loss, as well as reduced grain quality and germinability[1]. Improving seed dormancy would inhibit PHS and increase seed longevity. However, breeders often lack genetic resources to enhance seed dormancy, especially for important food crops, such as rice and wheat.

Primary seed dormancy is established during seed maturation. This quiescent state of seeds can be gradually broken by after-ripening (dry storage)[2]. Many internal and external physiological factors, such as phytohormones, temperature, light, and air humidity, affect seed dormancy. Among the dormancy-affecting factors, the phytohormones abscisic acid (ABA) and gibberellins (GAs) are two pivotal regulators that act antagonistically on seed dormancy[2–4]. Many other factors regulate seed dormancy through the ABA and/or GA pathways[2–4]. ABA positively regulates seed dormancy. As seeds mature, ABA level gradually increases to establish seed dormancy, and at the end of seed maturation, high ABA level prevents germination before harvest[3]. In contrast, GAs promote the transition from dormancy to germination and antagonize the effects of ABA[3,5,6]. Thus, the balance between these two phytohormones is critical in determining the status of seed dormancy[3]. Although major progress at physiological level has been made, our knowledge of the molecular mechanisms underlying seed dormancy is far from complete, particularly for crops.

In crops, the microRNA miR156 regulates grain yield by modulating plant shoot architecture and grain size[7–13]. miR156 targets a group of SQUAMOSA-PROMOTER BINDING PROTEIN-LIKE (SPL) transcription factor genes. In rice, the quantitative trait locus (QTL) Ideal Plant Architecture 1 (IPA1), a miR156 target gene, encodes SPL14 and was recently described as a "green revolution" gene for grain yield improvement[14]. In IPA1, a C-to-A single nucleotide polymorphism (SNP) within the coding region was reported to relieve its repression by miR156, and thus confers an ideal plant architecture with fewer tillers, stronger culms, more grains per panicle, and larger grains[7]. miR156 also regulates grain size through controlling the expression of SPL16 and SPL13[10–12]. With expanding knowledge about miR156, the miR156/SPL module has been proposed to be a versatile toolbox to enhance agronomic traits[15]. However, a comprehensive study regarding the potential functional differentiation of the many genes in the MIR156 family is still lacking, and the molecular mechanisms underlying the various functions of miR156 remain to be revealed.

Although miR156 overexpression and knockdown were reported to affect seed germination in Arabidopsis and rice[16,17], it is unclear whether miR156 is indeed involved in seed dormancy regulation. Here, through gene mutagenesis by CRISPR/Cas9 in rice, we found that mutations in a MIR156 subfamily (MIR156a–MIR156c, MIR156k, and MIR156l) markedly enhance seed dormancy and suppress PHS, with negligible effects on shoot architecture and grain size. In contrast, mutations in other MIR156 genes (MIR156d–MIR156i) modify plant architecture and increase grain size, but have minimal effects on seed dormancy. We show that mir156 mutations enhance seed dormancy by suppressing the GA pathway through up-regulation of IPA1. Our research reveals in vivo associations of IPA1 with the promoters of many GA biosynthetic, signaling, and deactivating genes, suggesting that IPA1 mediates the effects of mir156 mutations by directly regulating

multiple genes in the GA pathway. These results provide an effective method to suppress PHS without compromising productivity, and will facilitate breeding elite crop varieties with ideal plant architectures.

## Results

**Group I *MIR156* genes control shoot architecture.** The rice genome contains 11 *MIR156* genes expressing twelve miR156 precursors (pre-miR156a to pre-miR156l), with pre-miR156h and pre-miR156j transcribed from the same gene[18] (designated *MIR156h* in this study). To knockout the eleven *MIR156s*, we constructed six CRISPR/Cas9 vectors (I–VI) that specifically target the genomic sites corresponding to the mature miR156s (Supplementary Fig. 1a, c). Vector I targets six neighboring genes (group I *MIR156* genes: *MIR156d–MIR156i*) on the phylogenetic tree, and vector II targets *MIR156f* and *MIR156g* (Supplementary Figs. 1a and 2). We used an *Agrobacterium*-mediated method to transform the vectors into Nipponbare, a *japonica* variety widely used in laboratories, and Xiushui 134 (XS134), an elite *japonica* cultivar widely cultivated by farmers in southeast China. Using vectors I and II, we obtained many mutant lines of *mir156fg*, *mir156dehi*, *mir156deghi*, and *mir156defghi* (Supplementary Data 1a, b).

Phenotyping of these mutants was conducted in paddy fields of Shanghai (China) and Hangzhou (China). During the entire seedling stage, *mir156fg* and *mir156dehi* were similar in size to the wild type (Supplementary Fig. 3a–c). *mir156deghi* and *mir156-defghi* also showed similar size to the wild type before two or three-leaf seedling stage (Supplementary Fig. 3a), but subsequently the seedlings of these two mutants exhibited slightly smaller statures than the wild type (Fig. 1a and Supplementary Figs. 3b, c and 4a). In addition, the leaf blades of *mir156dehi*, *mir156deghi*, and *mir156defghi* seedlings in Nipponbare background were often more erect than those of the wild type (Fig. 1a).

At the mature stage, *mir156dehi*, *mir156deghi*, and *mir156defghi* showed apparent changes in plant architecture with significantly fewer tillers when compared with the wild type (Fig. 1b, c and Supplementary Fig. 4b, c). From *mir156dehi* to *mir156defghi*, higher-order mutants had fewer tillers (Fig. 1c). In addition, at the mature stage, *mir156dehi*, *mir156deghi*, and *mir156defghi* were taller than the wild type and displayed increased culm diameters (Fig. 1d, e).

In Nipponbare background, *mir156fg* often showed very slightly fewer tillers than the wild type (Fig. 1c and Supplementary Table 1). In XS134 background, obvious differences in tiller number were not observed between *mir156fg* and the wild type in successive three planting years (Supplementary Fig. 4c). Besides tiller number, we did not observe obvious differences in shoot architecture between *mir156fg* and the wild type (Fig. 1b–e). Consistent with these observations, we did not detect obvious differences in miR156 abundance and the expression of miR156 target genes between wild-type and *mir156fg* seedling shoots (Supplementary Fig. 5a–l).

To obtain additional group I mutants, we crossed *mir156dehi* with *mir156fg*, and identified new combinatorial mutants including *mir156defghi*, *mir156defgh*, *mir156defgi*, *mir156dfgi*, *mir156efgi*, *mir156efgh*, *mir156eghi*, *mir156efg*, *mir156fgi*, *mir156-ghi*, and *mir156ei* from the segregating F2 and F3 populations in Nipponbare background (Supplementary Data 1a). Overall, among these group I mutants, higher-order mutants showed taller and stronger statures but fewer tillers (Supplementary Table 1).

Together, these results indicate that group I *MIR156* genes control shoot architecture in rice.

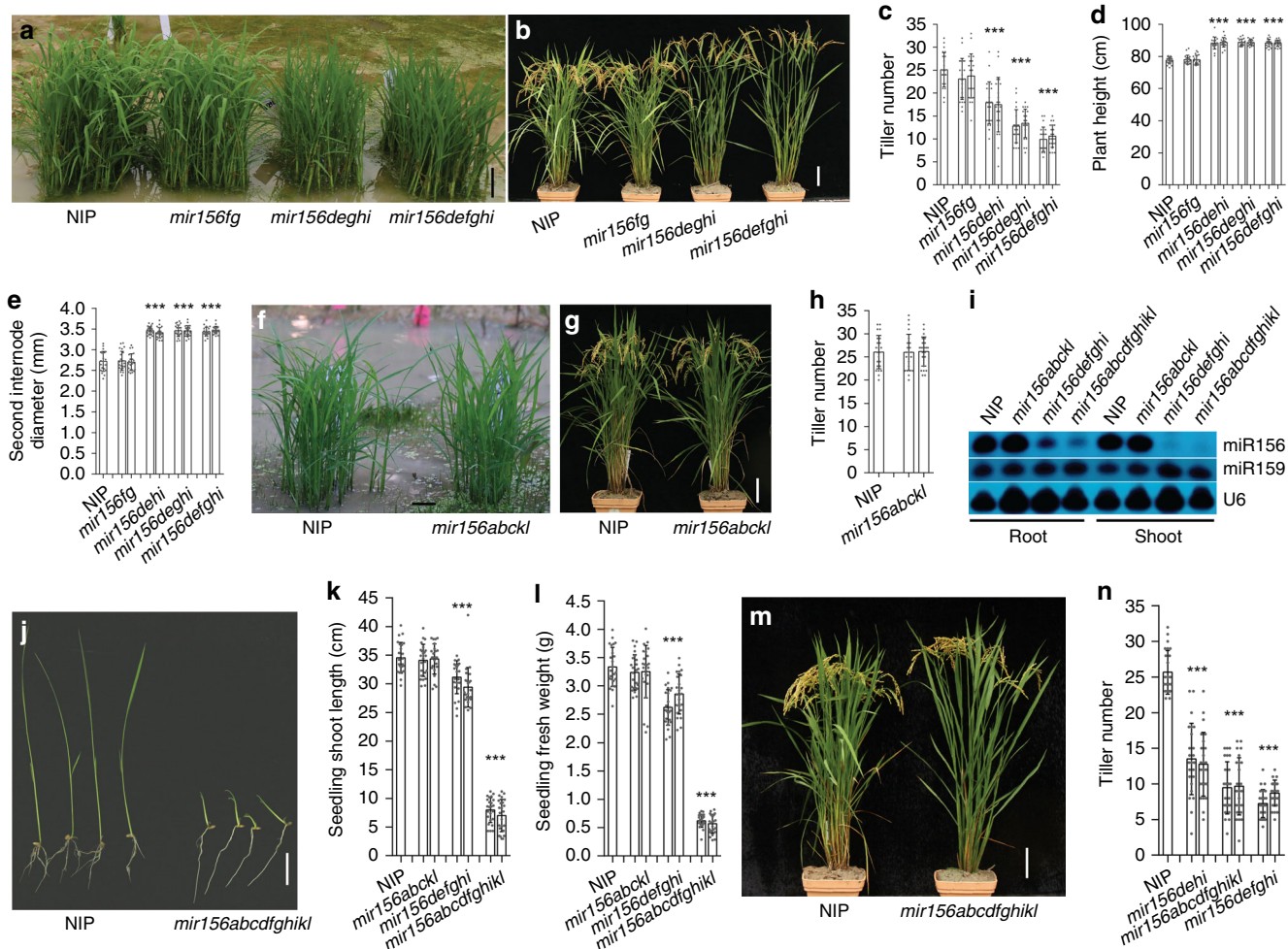

**Fig. 1** Shoot architectures of the *mir156* mutants. **a** 25-day-old seedlings of the wild type, *mir156fg*, *mir156deghi*, and *mir156defghi*. Scale bar, 5 cm. **b** Wild-type, *mir156fg*, *mir156deghi*, and *mir156defghi* plants at the mature stage. Scale bar, 10 cm. **c**, **d** Tiller numbers (**c**) and plant heights (**d**) of the wild type, *mir156fg*, *mir156dehi*, *mir156deghi*, and *mir156defghi* at the seed-filling stage. **e** Second internode diameters of wild-type, *mir156fg*, *mir156dehi*, *mir156deghi*, and *mir156defghi* main tillers. **f** 18-day-old seedlings of the wild type and *mir156abckl*. Scale bar, 3 cm. **g** Wild-type and *mir156abckl* plants at the mature stage. Scale bar, 10 cm. **h** Tiller numbers of the wild type and *mir156abckl* at the seed-filling stage. **i** Northern blot showing miR156 abundance in roots and shoots of 25-day-old seedlings. U6 RNA and miR159 were used as loading controls. **j** Ten-day-old seedlings of *mir156abcdfghikl* and the wild type. Scale bar, 3 cm. **k**, **l** Shoot lengths (**k**) and fresh weights (**l**) of 2-week-old wild-type, *mir156abckl*, *mir156defghi*, and *mir156abcdfghikl* seedlings. **m** Wild-type and *mir156abcdfghikl* plants at the mature stage. Scale bar, 10 cm. **n** Tiller numbers of the wild type, *mir156dehi*, *mir156abcdfghikl*, and *mir156defghi* at the seed-filling stage. Data are presented as means ± SD. Each bar in the bar charts represents an independent line. *P* values (versus the wild type) were calculated with Student's *t*-test. ***$P < 0.001$. NIP, Nipponbare. Source data are provided as a Source Data file

**Group II *MIR156s* have negligible effects on shoot architecture**. Using vectors III–VI (Supplementary Fig. 1a), we obtained many mutant lines of *mir156a*, *mir156b*, *mir156c*, *mir156ac*, *mir156bc*, *mir156abc*, *mir156kl*, *mir156k*, and *mir156l* (Supplementary Data 1a, b). No obvious differences in shoot architecture were observed between these mutants and the wild type in both Nipponbare and XS134 background.

To further study these five *MIR156* genes (*MIR156a*–*MIR156c*, *MIR156k*, and *MIR156l*), a multiplex gene-editing vector (vector VII) was constructed (Supplementary Fig. 1b) and transformed into Nipponbare and XS134 through an *Agrobacterium*-mediated method. Using this vector, we obtained five independent lines of *mir156abckl* in each background (XS134 and Nipponbare) (Supplementary Data 1a, b). All the *mir156abckl* lines showed smaller shoots than the wild type before two or three-leaf seedling stage (Supplementary Figs. 3d–g and 4d), but after this stage, no obvious morphological differences were observed between the wild type and *mir156abckl* (Fig. 1f–h and Supplementary Fig.

4e–g). We also carefully examined several other agronomic traits, including seed setting rate and grain number per main panicle, and did not observe obvious differences between *mir156abckl* and the wild type ($P > 0.45$) (Supplementary Fig. 6a–d). Consistently, the grain yield was not significantly affected by *mir156abckl* mutations ($P > 0.85$) (Supplementary Fig. 6e, f). Thus, *MIR156a*, *MIR156b*, *MIR156c*, *MIR156k* and *MIR156l* play minimal roles in regulating shoot architecture.

Consistent with the above results, miR156 expression was markedly decreased in *mir156defghi* seedlings (25-day-old at about five-leaf stage) but not in *mir156abckl* seedlings (25-day-old at about seven-leaf stage) (Fig. 1i). In addition, transcriptome analyses of the unelongated culms from 20-day-old seedlings revealed that miR156 target genes were not enormously up-regulated by *mir156abckl* mutations (ratio < 2), whereas several miR156 target genes (*SPL3*, *SPL13*, *IPA1*, and *SPL17*) were highly up-regulated by *mir156defghi* mutations (ratio > 2) (Supplementary Fig. 7a, b; Supplementary Tables 2 and 3). Thus, to

correspond with the "group I" designation for *MIR156d*–*MIR156i*, we designated *MIR156a*, *MIR156b*, *MIR156c*, *MIR156k*, and *MIR156l* as "group II" *MIR156* genes.

**Plant growth is suppressed in decuple *mir156* mutants**. We obtained two lines of decuple mutant *mir156abcdfghikl* via particle bombardment to co-transform vectors I–VI into Nipponbare (Supplementary Data 1a). Through the crossing between *mir156abckl* and *mir156defghi*, we also identified another *mir156abcdfghikl* line in Nipponbare background (Supplementary Data 1a). These three *mir156abcdfghikl* lines were similarly smaller than the wild type before the heading stage (Supplementary Fig. 8a–d), and showed severely retarded growth during the seedling stage (Fig. 1j–l and Supplementary Fig. 8b). However, at the mature stage, *mir156abcdfghikl* plants were taller than the wild type (Fig. 1m and Supplementary Fig. 8e). In the paddy field, *mir156abcdfghikl* tiller number was slightly larger than that of *mir156defghi*, but smaller than that of *mir156dehi* (Fig. 1n).

We also attempted to obtain miR156 knockout mutants (*mir156abcdefghikl*) by crossing *mir156abcdfghikl* with *mir156defghi*, and two heterozygous *mir156abcdefghikl* plants, in which only *MIR156k* was heterozygous (+/−), were identified in F4 generation. In F5 and F6 generations, most offspring of the heterozygous *mir156abcdefghikl* died during the seedling stage (more severely than *mir156abcdfghikl* seedlings), and the remaining plants were homozygous *mir156abcdefghil* or heterozygous *mir156abcdefghikl*. Among the dead seedlings, we identified homozygous *mir156abcdefghikl* plants (miR156 knockout plants), suggesting that miR156 is essential for survival in rice. Compared with the wild type and *mir156defghi*, the homozygous *mir156abcdefghil* showed dwarf stature with significantly fewer tillers (Supplementary Fig. 9a, b). Thus, group I and II *MIR156s* act redundantly to support plant growth, especially at the seedling stage.

**Group I *MIR156s* regulate grain size**. Three targets of miR156 (*IPA1*, *SPL16*, and *SPL13*) positively regulate grain size[7,10–12]. Therefore, we examined the grain size and shape of the *mir156* mutants. We found that the grain lengths and 1000-grain weights of *mir156fg*, *mir156dehi*, *mir156defghi* and *mir156abcdfghikl* were significantly greater than those of the wild type (Fig. 2a, b, e and Supplementary Fig. 4h, i, l). No significant differences in grain width and thickness were observed between the wild type and all *mir156* mutants (Fig. 2c, d and Supplementary Fig. 4j, k). The grain size and shape of *mir156abckl* were similar with those of the wild type (Fig. 2b–e and Supplementary Fig. 4i–l). Moreover, *mir156abcdfghikl* did not show larger grain size than *mir156defghi* (Fig. 2b–e).

These results indicate that group I *mir156* mutations increase grain size, whereas the group II genes play negligible roles in modulating grain size and shape. In addition, the above results also suggest that *mir156fg* is a valuable genetic resource for grain yield improvement. *mir156fg* mutations significantly increased grain size (Fig. 2a, b, e and Supplementary Fig. 4h, i, l) but had minimal effects on shoot architecture (Fig. 1b–e). Besides grain size and shoot architecture, other agronomic traits were not obviously affected by *mir156fg* mutations.

**mir156 mutations suppress adventitious root formation**. In *Arabidopsis*, tobacco, and *Malus xiaojinensis*, miR156 is required for adventitious or lateral root formation[19,20]. Therefore, the root development in the *mir156* mutants was investigated. We found that in 20-day-old seedlings, only 1–2 adventitious roots were formed in the decuple mutant *mir156abcdfghikl*, whereas over forty adventitious roots appeared in the wild type (Supplementary

Fig. 10a, c). *mir156defghi* and *mir156abckl* seedlings also showed fewer adventitious roots than the wild type (Supplementary Fig. 10b–d). Thus, both group I and group II *MIR156s* are required for adventitious root formation.

**Group II *MIR156s* regulate seed dormancy**. At the sowing time every year in paddy fields, we observed obviously slower seed germination in *mir156abc*, *mir156abcl*, *mir156abckl* and *mir156abcdfghikl* than in the wild type. The slow germination was not clearly observed in group I mutants. Germination assays confirmed the slower germination in *mir156abcdfghikl* and group II *mir156* mutants (Fig. 3a, b and Supplementary Fig. 11a, b). Consistent with these results, miR156 expression was obviously decreased in the germinating embryos of *mir156abckl* and *mir156abcdfghikl* (Supplementary Fig. 12a). These observations indicate that miR156 may negatively regulate seed dormancy in rice.

To test this hypothesis, we conducted seed germination assays using fresh wet seeds (freshly harvested seeds without drying treatment, and termed fresh seeds hereinafter). We found that *mir156abcdfghikl* fresh seeds hardly germinated after culturing for more than 2 weeks (Fig. 3d), and the group II mutants, including *mir156abc*, *mir156abcl*, and *mir156abckl*, showed obviously slower germination than the wild type and *mir156defghi*, with higher-order mutants exhibiting more delayed germination (Fig. 3c, d and Supplementary Fig. 11c, d). In this assay, *mir156defghi* showed only a slightly slower germination than the wild type (Fig. 3c, d). Thus, group II *MIR156s* play more important roles than the group I in regulating seed dormancy. Consistently, expression analyses revealed that several miR156 target genes (*SPL12*, *SPL13*, and *IPA1*) were more intensely up-regulated by *mir156abckl* mutations than by *mir156defghi* mutations (Supplementary Fig. 12b–k).

**Group II *MIR156s* control PHS and seed longevity**. In crops, PHS frequently results in severe loss of grain yield and quality. Enhanced seed dormancy is expected to prevent PHS. To explore whether *mir156* mutations suppress PHS, we investigated the PHS rates of the wild type and *mir156* mutants in Nipponbare background in the years 2017 and 2018. In southeast China including Hangzhou, the wet weather during harvest time (late September) often induces severe PHS in Nipponbare. In Hangzhou paddy field, we observed significantly lower PHS rates in *mir156abcl* and *mir156abckl* than in *mir156defghi* and wild-type Nipponbare at the normal harvest time (Fig. 3e and Supplementary Fig. 13). *mir156defghi* showed only slightly lower PHS rates than the wild type (Fig. 3e and Supplementary Fig. 13). These results indicate that group II *mir156* mutations markedly suppress PHS.

Enhanced seed dormancy is also expected to extend seed viability. Therefore, we conducted germination assays using seeds stored for different times. We found that *mir156abckl* lines had significantly higher germination rates than the wild type after storage for 14 months (Fig. 3f), indicating that *mir156abckl* mutations are helpful in maintaining seed viability over time.

**ABA does not support the enhanced seed dormancy in *mir156s***. Because ABA positively regulates seed dormancy, we measured ABA level in fresh seeds of the wild type and *mir156* mutants. The results showed that compared to the wild type, *mir156abcdfghikl* contained severely a decreased level of ABA in the fresh seed embryos (Fig. 4a). ABA level was also significantly decreased in the fresh seeds of *mir156abckl* (Fig. 4b). Thus, enhancing seed dormancy by *mir156* mutations is not through increased accumulation of ABA.

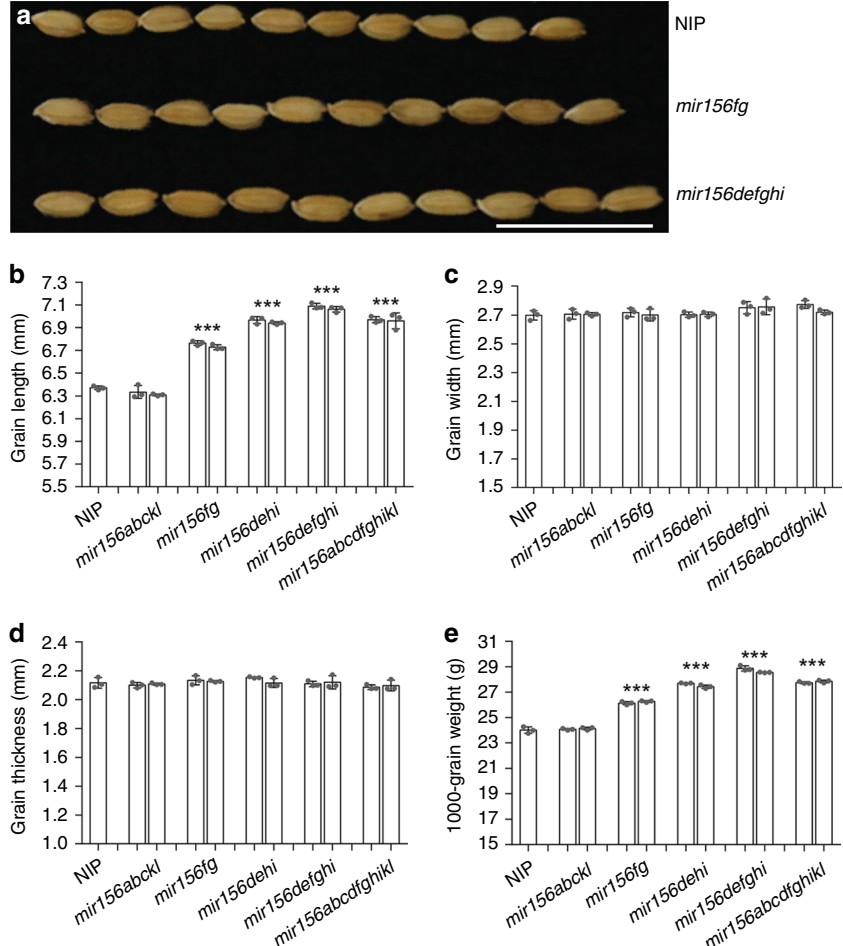

**Fig. 2** Grain shape analyses of the *mir156* mutants. **a** Seeds of the wild type, *mir156fg* and *mir156defghi*. Scale bar, 2 cm. **b–d** Grain lengths (**b**), widths (**c**), and thicknesses (**d**) of the wild type, *mir156abckl*, *mir156fg*, *mir156dehi*, *mir156defghi*, and *mir156abcdfghikl*. **e** 1000-grain weights of the wild type, *mir156abckl*, *mir156fg*, *mir156dehi*, *mir156defghi*, and *mir156abcdfghikl*. Data are presented as means ± SD. Each bar in the bar charts represents an independent line. *P* values (versus the wild type) were calculated with Student's *t*-test. ***$P < 0.001$. NIP, Nipponbare. Source data are provided as a Source Data file

ABA receptors are encoded by the *Pyrabactin Resistance 1-like* (*PYL*) family genes[21]. It was previously reported that mutations in a sub-family of rice *PYLs* led to severe defects in seed dormancy[22]. To explore a potential relationship between miR156 and the ABA pathway in regulating seed dormancy, we crossed *mir156abckl* with *pyl1/4/6* and *pyl1/2/3/4/5/6/12*, and identified *pyl1/4/6-mir156abckl* and *pyl1/2/4/6-mir156abckl* in the segregating F3 populations. Germination assays with fresh seeds showed that *mir156abckl* mutations markedly suppressed the seed dormancy defects in *pyl1/4/6* and *pyl1/2/4/6*, and the seed dormancy of *pyl1/4/6-mir156abckl* was even stronger than that of the wild type (Fig. 4c, d and Supplementary Fig. 14a, b).

The above results suggest that the enhanced seed dormancy in *mir156* mutants is not mediated by the ABA pathway.

**mir156 mutations enhance seed dormancy via the GA pathway.** To uncover the mechanisms underlying the enhanced seed dormancy in *mir156* mutants, we analyzed the transcriptomes of *mir156abcdfghikl* and wild-type fresh seed embryos (Supplementary Data 2). A total of 1132 differentially expressed genes (DEGs) (ratio ≥ 2 or ≤ 0.5, and false discovery rate (FDR) < 0.05), including 729 up-regulated and 403 down-regulated genes in *mir156abcdfghijkl* compared to the wild type, were identified (Supplementary Data 3). Among the DEGs, two putative GA receptor genes (LOC_Os06g20200 and LOC_Os09g28630), three

key GA biosynthetic genes (*GNP1*, *SD1*, and *KAO*)[23], and a putative GA oxidase gene (LOC_Os03g42130) were significantly down-regulated in *mir156abcdfghikl* (Supplementary Data 4). Moreover, a GA deactivating DEG *gibberellin 2-oxidase 6* (*GA2ox6*)[24] was markedly up-regulated in *mir156abcdfghikl* (Supplementary Data 4). When lowering the threshold for the DEGs to ratio ≥ 1.5 or ≤ 0.75 (FDR < 0.05), we found down-regulation of another three putative GA receptor genes (LOC_Os08g37040, LOC_Os02g35940, and LOC_Os09g28730) and up-regulation of another two GA deactivating genes (*GA2ox8*[24] and *ELONGATED UPPERMOST INTERNODE 1* (*EUI1*)[25,26]) in *mir156abcdfghikl* (Supplementary Data 4). These results were validated by quantitative real-time RT-PCR (RT-qPCR) (Fig. 5a). In *mir156abckl* fresh seed embryos, RT-qPCR also showed decreased expression of the GA biosynthetic and receptor genes but increased expression of the GA deactivating genes (Supplementary Fig. 15).

Next, we measured GA levels in the fresh seed embryos. The results revealed that *mir156abcdfghikl* accumulated much lower levels of two bioactive GAs, $GA_3$, and $GA_7$, than the wild type (Fig. 5b). We could not detect other bioactive GAs ($GA_1$ and $GA_4$) in the fresh seed embryos of both *mir156abcdfghijkl* and the wild type. Furthermore, the sensitivity of seed germination to GAs was tested using fresh seeds. $GA_3$-promoted germination was clearly observed in the wild type during the entire observing time, while the effect of $GA_3$ on the germination of *mir156abckl*

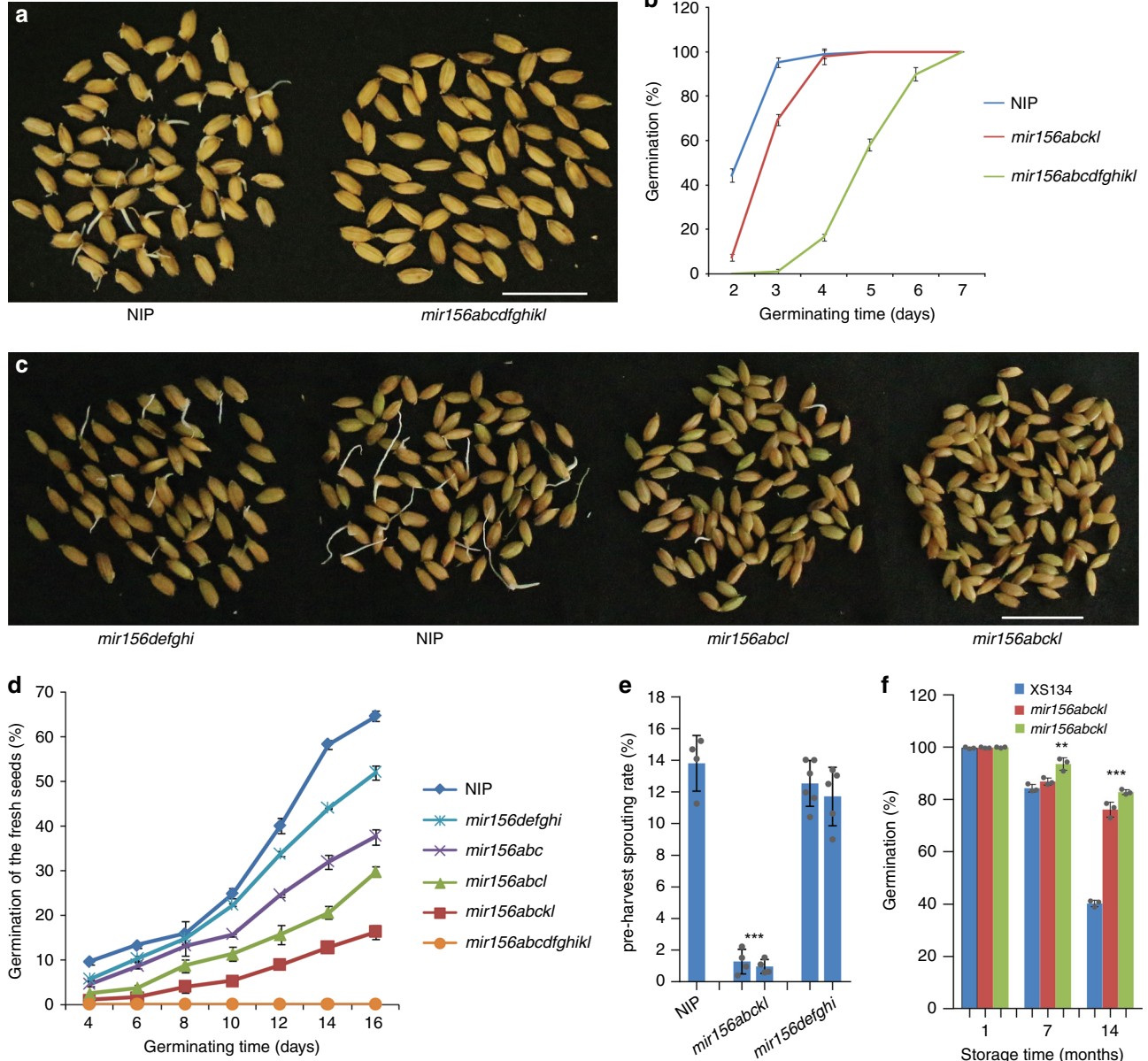

**Fig. 3** *mir156* mutations enhanced seed dormancy. **a**, **b** Seed germination comparison of the wild type, *mir156abckl* and *mir156abcdfghikl*. After harvest, the seeds for this assay were immediately dried in a 42 °C dry oven for 7 days, and then stored in a 20 °C dry cabinet for 2 weeks. Scale bar, 2 cm. **c**, **d** Fresh seed germination comparison of the wild type, *mir156abc*, *mir156abcl*, *mir156abckl*, *mir156defghi*, and *mir156abcdfghikl*. Scale bar, 2 cm. **e** PHS rates of wild-type, *mir156abckl* and *mir156defghi* seeds in Hangzhou in the year 2018. **f** Germination rates of wild-type and *mir156abckl* seeds after storage for the indicated times at room condition in Hangzhou. Each bar in the bar charts represents an independent line. Data are presented as means ± SD. *P* values (versus the wild type) were calculated with Student's *t*-test. ***$P < 0.001$. NIP, Nipponbare. Source data are provided as a Source Data file

seeds was not clearly observed in the first 5 days, and only from the sixth day of the treatment, the effect of $GA_3$ on the germination of *mir156abckl* seeds became evident (Fig. 5c), indicating less sensitivity of seed germination to $GA_3$ in *mir156abckl* than in the wild type.

Altogether, these results support that *mir156* mutations enhance seed dormancy through suppressing GA biosynthesis and signaling but promoting GA deactivating.

In addition to enhancing seed dormancy, *mir156* mutations also suppress seedling growth. The morphological phenotype of *mir156abcdfghikl* seedlings (Fig. 1j and Supplementary Fig. 8b) closely resembles those of some GA-deficient mutants, such as *GA-insensitive dwarf 1* (*gid1*)[27], *GA-insensitive dwarf 2* (*gid2*)[28], and strong alleles of *SLENDER RICE 1* (*SLR1*) gain-of-function

mutants[29]. Consistent with this severely retarded seedling growth, *mir156abcdfghikl* seedling shoots contained much lower levels of the bioactive GAs including $GA_3$, $GA_4$, and $GA_7$ than the wild type (Fig. 5d). In addition, transcriptome analyses (Supplementary Data 5) revealed that most of the GA biosynthetic and signaling genes were down-regulated in *mir156abcdfghikl* seedling shoots compared to the wild type (Supplementary Fig. 16; Supplementary Data 6). Among the DEGs (ratio ≥ 2 or ≤ 0.5, and FDR < 0.05) identified in *mir156abcdfghikl* and wild-type seedling shoots, 14 putative GA receptor genes and five key GA biosynthetic genes (*CPS1*, *KAO*, *KO2*, *GNP1*, and *GA20ox4*)[23] were significantly down-regulated in *mir156abcdfghikl*, whereas a GA deactivating gene *GA2ox10*[24] was markedly up-regulated in *mir156abcdfghikl* (Supplementary Data 7). Through the

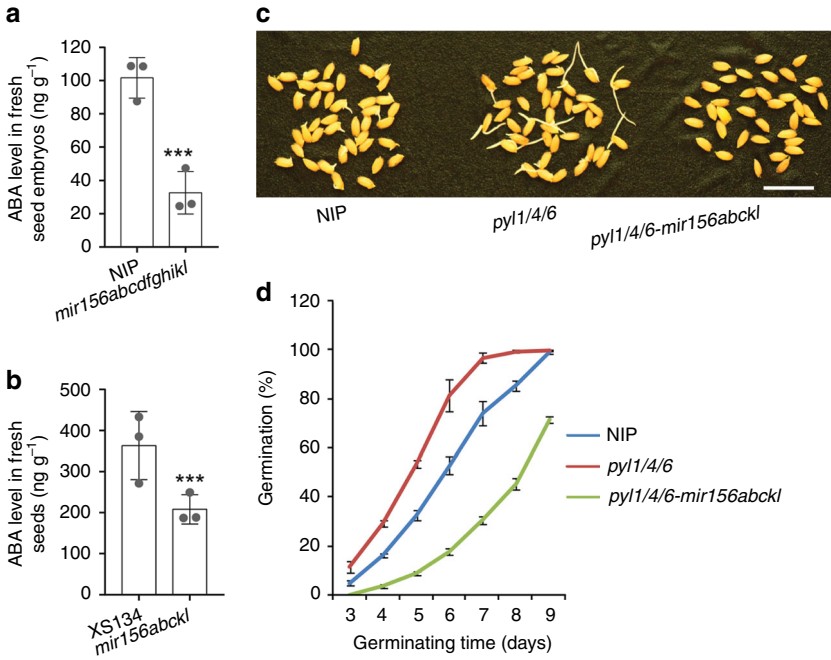

**Fig. 4** Enhanced seed dormancy in *mir156* mutants is not mediated by the ABA pathway. **a** ABA level in wild-type and *mir156abcdfghikl* fresh seed embryos. **b** ABA level in wild-type and *mir156abckl* fresh seeds. **c**, **d** Fresh seed germination comparison of the wild type, *pyl1/4/6* and *pyl1/4/6-mir156abckl*. Scale bar, 2 cm. Data are presented as means ± SD. *P* values (versus the wild type) were calculated with Student's *t*-test. ***$P < 0.001$. NIP, Nipponbare. Source data are provided as a Source Data file

transcriptome analyses, we also found that *SLR1* and *SPINDLY* (*SPY*), two negative regulators of GA signaling[29–31], were up-regulated in *mir156abcdfghikl* seedling shoots compared to the wild type ($1.5 < \text{ratio} < 2$) (Supplementary Data 6). We confirmed these results by RT-qPCR (Fig. 5e). Consistent with the increased *SLR1* expression in *mir156abcdfghikl*, Western blotting assays suggest slightly increased accumulation of SLR1 protein in *mir156abcdfghikl* seedling shoots compared to the wild type (Supplementary Fig. 17).

Overall, these results strongly support that *mir156* mutations enhance seed dormancy and inhibit seedling growth through suppressing the GA pathway.

**Overexpression of the miR156 target gene *IPA1***. In rice, miR156 acts through targeting 11 *SPL* genes (*SPL2–SPL4*, *SPL7*, *SPL11–SPL13*, *IPA1/SPL14*, and *SPL16–SPL18*)[18]. Our transcriptome analyses revealed that among the miR156 target genes, *IPA1* was expressed most highly in both *mir156abcdfghikl* and wild-type fresh seed embryos, and especially in *mir156abcdfghikl* fresh seed embryos, *IPA1* expression level was extremely higher than those of other miR156 target genes (Fig. 6a and Supplementary Table 4). This result suggests that *mir156abcdfghikl* mutations enhance seed dormancy mainly through increasing *IPA1* expression. The transcriptome analyses also revealed that among the miR156 target genes, *IPA1* was up-regulated most intensely by *mir156abcdfghikl* mutations in the seedling shoots (Supplementary Table 5; Supplementary Fig. 18), suggesting that *IPA1* also mediates the effects of *mir156abcdfghikl* mutations on seedling growth. Therefore, we examined whether *IPA1* over-expression affected seed dormancy and plant growth. *IPA1* overexpression driven by the seed-specific *Ole18* promoter[32] resulted in significantly enhanced seed dormancy (Supplementary Fig. 19a–c) without noticeable effects on plant architecture. In addition, *IPA1* overexpression driven by the cauliflower mosaic virus 35S promoter resulted in not only significantly enhanced seed dormancy but also markedly smaller plant statures

compared to the wild type (Fig. 6b–g and Supplementary Fig. 8d, e). These results indicate that *IPA1* mediates the effects of *mir156* mutations on seed dormancy and plant growth.

**Associations of IPA1 with the promoters of GA-related genes**. Since IPA1 is a transcription factor[33], it is possible that IPA1 directly regulates GA-related genes. Therefore, we searched previously published ChIP-seq data of IPA1[33], and noticed that seven GA-related genes, including *EUI1*, *GA2ox6*, *SLR1*, *GSR1* (*GAST family gene in rice 1*, a GA-stimulated transcription factor[34]), *GID2*, and two putative GA receptor genes (LOC_Os02g35940 and LOC_Os06g20200), were among the genes with IPA1 binding sites in the 500-bp promoter regions or first introns. Furthermore, IPA1 binding on *SLR1* promoter has been confirmed by gel electrophoresis mobility shift assays (EMSAs)[33]. The biological significance of the interactions between IPA1 and these GA-related genes remains to be revealed.

SPL family proteins, including IPA1, directly bind to the GTAC motif to regulate gene expression[33]. In rice, IPA1 also binds to the TGGGCC/T motif via interacting with PCF1 and PCF2[33]. Through sequence analyses of 1300-bp promoter regions, we found that most of the identified GA-related DEGs contained multiple GTAC and/or TGGGCC/T motifs. Therefore, we used ChIP-qPCR to explore associations between IPA1 and 16 of the GA-related DEGs identified above. In $P_{35S}$:*IPA1m:3×FLAG* transgenic seedling shoots (see Supplementary Fig. 20a for *IPA1* expression in the $P_{35S}$:*IPA1m:3×FLAG* lines), the ChIP-qPCR revealed in vivo interactions between IPA1 and the promoters of 14 GA-related DEGs (Fig. 7a–e and Supplementary Fig. 20b–j). The 14 DEGs included five putative GA receptor genes (LOC_Os09g28630, LOC_Os06g20200, LOC_Os02g35940, LOC_Os08g37040, and LOC_Os09g28730), five key GA biosynthetic genes (*CPS1*, *KAO*, *KO2*, *GNP1*, and *SD1*), three GA deactivating genes (*GA2ox6*, *GA2ox8*, and *EUI1*), and *SLR1*. Subsequently, EMSA assays confirmed that IPA1 directly bound to the GTAC-containing fragments in the promoters of *KAO*,

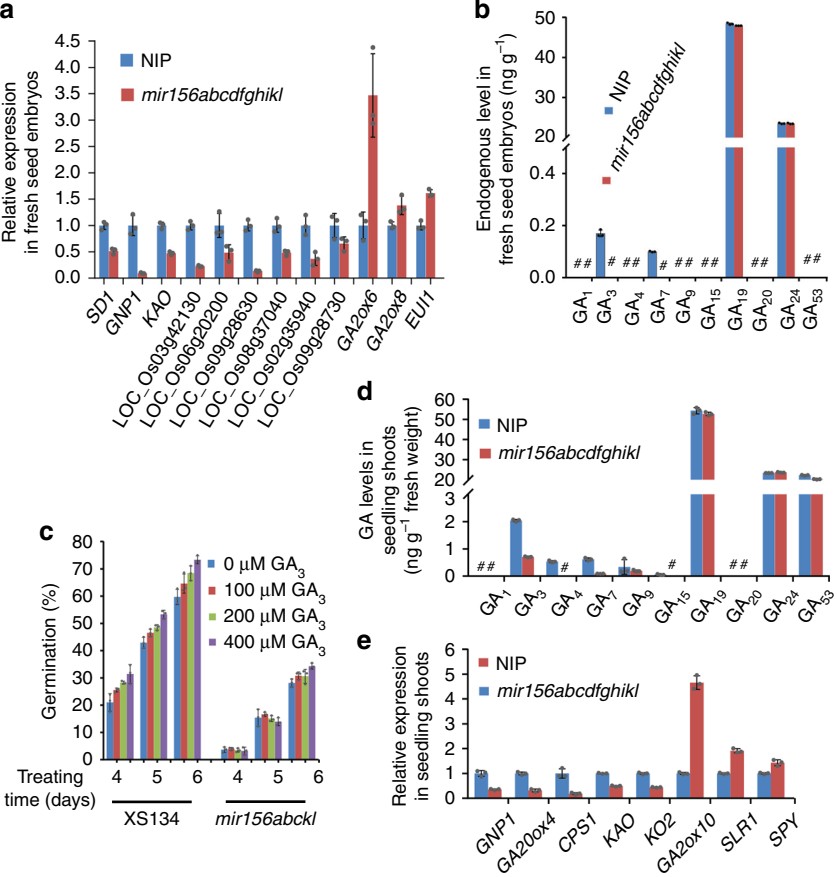

**Fig. 5** *mir156* mutations enhanced seed dormancy and suppressed seedling growth by suppressing the GA pathway. **a** Relative expression levels of the GA biosynthetic, signaling and deactivating DEGs in fresh seed embryos. **b** Endogenous GA levels in fresh seed embryos. #, undetectable GAs. **c** Sensitivity of fresh seed germination to GA₃. **d** Endogenous GA levels in the seedling shoots of two-week-old plants. #, undetectable GAs. **e** Relative expression levels of several GA-related DEGs in the seedling shoots of 2-week-old plants. Three independent biological replicates were performed, and error bars indicate standard deviation. NIP, Nipponbare. Source data are provided as a Source Data file

*CPS1, GNP1, SD1, SLR1, GA2ox6*, and *EUI1* (Supplementary Fig. 21a–h). These results suggest that IPA1 directly regulates multiple GA biosynthetic, signaling, and deactivating genes. In the ChIP-qPCR assays, we did not detect interactions between IPA1 and the promoters of the other two DEGs, *GA20ox4* and *GA2ox10*. These two DEGs may be directly regulated by SPL proteins encoded by other miR156 target genes, or may be indirectly regulated by IPA1.

Overall, our results strongly support that *mir156* mutations enhance seed dormancy and inhibit seedling growth by suppressing the GA pathway, and IPA1 mediates these effects of *mir156* mutations largely by directly regulating multiple genes in the GA pathway.

## Discussion

In recent years, with expanding knowledge about miR156, the miR156/SPL module has been proposed as a versatile toolbox to enhance agronomic traits[15]. In crop breeding practices, utilizations of the QTLs *IPA1, GW8* (*SPL16*) and *GLW7* (*SPL13*), which modify the regulatory circuit of the miR156/SPL module, have greatly improved grain yield through modulating plant architecture and grain size[15]. Although the miR156/SPL module is powerful in regulating plant architecture and grain size, its pleiotropic effects (effects on tillering, tiller angle, plant height, grain shape, etc) often become obstacles for its applications. In addition, miR156 knockdown or the QTL *ipa1* leads to extreme

changes in plant architecture[7,8,35,36], whereas crop breeders often need to change plant architecture to less extent. Therefore, breeders are often not able to use the miR156/SPL module to adjust plant architecture at will. In this study, through systematic gene editing, we revealed the functional differentiation of *MIR156s* in detail, and these results will enable breeders to adjust plant architecture and grain size according to the dosage they need and without undesirable pleiotropic effects (through combinatorial gene editing of group I *MIR156s*).

In crops, PHS frequently leads to severe loss of grain yield and quality. Enhanced seed dormancy would inhibit PHS. However, the genes known to affect seed dormancy without penalty on productivity are limited. Only two seed dormancy-controlling QTL genes have been identified in crops, including *Seed dormancy 4* (*Sdr4*) in rice and *Qsd1* in barley[37,38]. In bread wheat, a *mitogen-activated protein kinase kinase 3* (*MKK3*) gene was recently proposed as a candidate gene for the seed dormancy-controlling QTL *Phs1*[39]. Utilizing these few QTL genes requires time-consuming efforts in crossing between different varieties, which may introduce undesirable agronomic traits. No other genes have been reported to mainly affect seed dormancy in crops. ABA positively regulates seed dormancy. Although manipulating the genes in the ABA pathway could enhance seed dormancy, this process would also cause suppression in plant growth and thus reduction in productivity.

In this study, we made the exciting observation that disrupting group II *MIR156s* (*MIR156a*–*MIR156c*, *MIR156k*, and *MIR156l*)

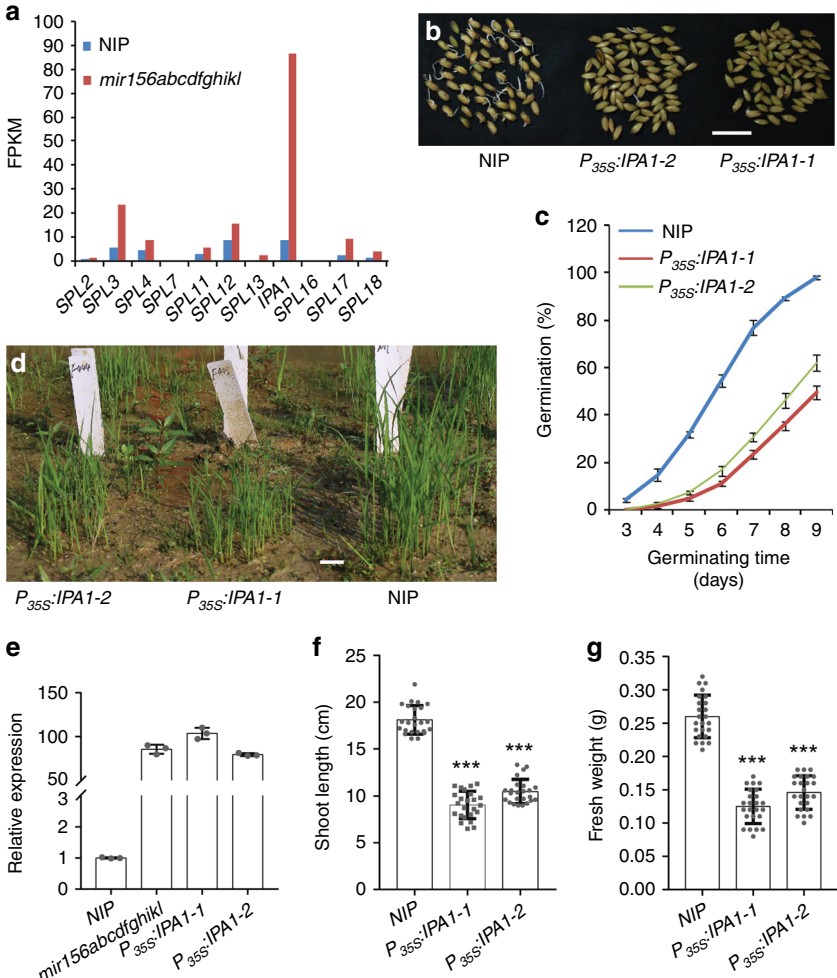

**Fig. 6** *IPA1* overexpression enhanced seed dormancy and suppressed seedling growth. **a** Relative expression levels of miR156 target genes in the fresh seed embryos. FPKM, fragments per kilobase of exon per million reads mapped. The data were obtained from the transcriptome analyses. The FPKM values were used to compare relative level of gene expression. **b, c** Fresh seed germination comparison of the wild type and two *P35S:IPA1* overexpression lines. Scale bar, 2 cm. **d** Two-week-old seedlings of the wild type and two *P35S:IPA1* lines. Scale bar, 3 cm. **e** Relative expression levels of *IPA1* in wild-type, *mir156abcdfghikl*, and *P35S:IPA1* seedling shoots. Two-week-old seedlings were used in this assay. **f, g** Shoot lengths (**f**) and fresh weights (**g**) of 2-week-old wild-type and *P35S:IPA1* seedlings. Data are presented as means ± SD. *P* values (versus the wild type) were calculated with Student's *t*-test. ***$P < 0.001$. NIP, Nipponbare. Source data are provided as a Source Data file

markedly enhanced seed dormancy and inhibited PHS with negligible effects on shoot architecture and grain size. Our results also showed that *mir156abckl* mutations helped to maintain seed viability for longer time. Yield tests in paddy fields showed that *mir156abckl* mutations did not obviously affect grain yield. Therefore, through *MIR156* gene mutagenesis by CRISPR/Cas9, we provided an efficient and effective method to inhibit PHS and increase seed longevity without compromising productivity.

In addition, according to our results, through combinatorial gene editing of group I and group II *MIR156s*, rice breeders should be able to improve plant architecture, grain size, and seed dormancy simultaneously, and thus generate PHS-resistant crops with increased grain productivity. Given the high conservation of miR156 in plants, this strategy may be also applicable to other crops.

Furthermore, our work shows that miR156 regulates seed dormancy and seedling growth through a molecular network composed of *IPA1* and many downstream genes in the GA pathway. Considering the similarity between IPA1 and other miR156-targeting SPLs, as well as mis-regulations of these *SPL* genes in *mir156* mutants (Fig. 6a and Supplementary Table 5), it

is plausible that miR156 also regulates the GA pathway through other miR156 target genes (*SPL2–SPL4, SPL7, SPL11–SPL13*, and *SPL16–SPL18*). Thus, our study should prompt the readers of our work to speculate that miR156 regulates seed dormancy and plant growth through a complex molecular network which composes of *SPLs* (including *IPA1*) and many downstream genes in the GA pathway.

ABA is an important stress phytohormone that enhances plant adaptation to abiotic and biotic stresses[22]. In *mir156abcdfghikl* and *mir156abckl*, we observed decreased ABA levels in seeds. The decreased ABA levels may be a response to the impaired GA pathway in *mir156* mutants. Although the ABA level was reduced in seeds, stress resistance does not appear to be compromised in *mir156* mutants. Deficiency in GA pathway can improve abiotic stress resistance[40]. In *mir156* mutants, impaired GA pathway was observed. Consistent with the impaired GA pathway, we found that group I *mir156* mutations reduced transpirational water loss rates of detached flag leaf blades and improved drought stress tolerance of the seedlings (Supplementary Figs. 22 and 23). In addition, it was previously reported that miR156 knockdown or *IPA1* overexpression conferred resistance against rice blast and

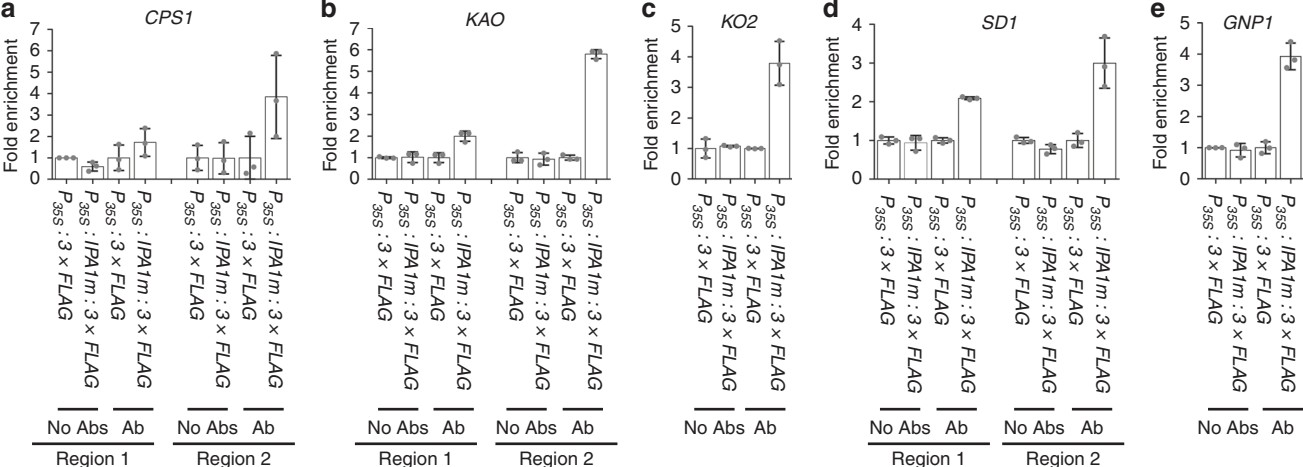

**Fig. 7** ChIP-qPCR assays showing in vivo interactions between IPA1 and promoters of five key GA biosynthetic genes. **a–e** Interactions between IPA1 and promoters of *CPS1* (**a**), *KAO* (**b**), *KO2* (**c**), *SD1* (**d**), and *GNP1* (**e**). IPA1m, IPA1-coding sequence with synonymous mutations at miR156-targeting site; NoAbs, without antibodies; Ab, antibodies against FLAG; regions, the fragments in the promoters detected for interactions with IPA1. Values are means ± SE (*n* = 3). The fold enrichment was normalized against the promoter of *UBIQUITIN*. Source data are provided as a Source Data file

bacterial blight, two major devastating diseases in rice[17,41]. Thus, *MIR156* knockout through gene editing may improve both agronomic traits and stress resistance simultaneously.

## Methods

**Plant material construction and cultivation**. Six single sgRNA-expressing vectors (vector I–vector VI) were constructed for editing the 11 *MIR156* genes (Supplementary Fig. 1a). To generate *mir156abckl*, a multiplex gene-editing vector (vector VII) was constructed, and in this vector, four sgRNA-expression cassettes were arranged in tandem (Supplementary Fig. 1b). The sgRNAs used in this study were designed to specifically target *MIR156* genes. Sanger sequencing was performed to identify *mir156* mutants from the transgenic plants. Due to the high sequence similarity between miR156 members, the sgRNAs designed for certain *MIR156* genes may also induce off-target mutations in other *MIR156* genes. Therefore, we sequenced both the target sites and potential off-target sites in *MIR156* genes to identify the genotype. In addition, to accurately ascertain the genotype and heritability of every line, we sequenced the target genes in at least two generations and selected T-DNA-free plants for seed harvesting.

In *P_{Ole18}:IPA1* and *P_{35S}:IPA1*, the IPA1-coding sequence was placed under the controls of *Ole18* and 35S promoters, respectively, in the pCAMBIA1300 backbone. Through *Agrobacterium*-mediated transformation, more than thirty transgenic lines were generated for each type of overexpression (*P_{35S}:IPA1* and *P_{Ole18}:IPA1*). The transgenic plants were grown in the paddy field. The seeds from the transgenic plants were sowed in Shanghai (China) and Hangzhou (China) in early June.

**Seedling investigations**. Seeds having just germinated at similar times were transferred to one-half-strength Kimura B liquid media and cultured in greenhouse (26 °C, 80% humidity, and 12 h light/12 h dark). Then the shoot lengths, fresh weights, and adventitious root numbers of the seedlings were investigated at indicated times. For investigation in paddy field, seeds of the wild type and *mir156* mutants were sowed in the field at similar times, and then the seedlings were investigated every day.

**Plot grain yield test**. Plants of the wild type and *mir156abckl* were grown in Hangzhou paddy field under natural conditions. The area per plot was 90 cm × 60 cm, and 24 plants were cultivated in each plot with planting density of 15 cm × 15 cm.

**Small RNA Northern blot**. Small RNA Northern blots were performed based on a previous report[42]. In brief, 20 μg total RNAs from the indicated materials were separated by running a 15% PAGE with 7 M urea, and then transferred to a Hybond NX membrane (GE, Amersham). After that, a 1-ethyl-3-(3-dimethylaminopropyl) carbodiimide (EDC) mediated chemical crosslinking was carried out. Antisense complementary oligonucleotides for miRNAs and U6 probes were end labeled ([γ-$^{32}$P] ATP) by T4 polynucleotide kinase (*New England Biolabs*). The probe sequences are listed in Supplementary Table 6.

**Seed dormancy comparison and germination assay**. In Hangzhou and Shanghai, the heading date was delayed by about 2, 9, and 5–12 days by *mir156defghi*, *mir156abcdfghikl*, and *P_{35S}:IPA1* overexpression, respectively. Other mutants had

heading dates comparable with the wild type in these two places. Therefore, before seed dormancy comparison, to assure comparability between the fresh seeds of different materials, *mir156defghi*, *mir156abcdfghikl* and *P_{35S}:IPA1* lines were planted earlier than the wild type and other mutants to ensure all the materials to flower at similar dates. For the seed dormancy comparison, fresh seeds were first soaked in water for two days, and then continued to germinate at 30 °C. For other germination assays, the seeds were first soaked in water for one day, and then continued to germinate at 30 °C. These assays were repeated three times, and in every repeat, at least 100 seeds of every line were used.

**PHS investigation**. PHS was investigated in the paddy field of Hangzhou at the normal harvest time. In every line, all of the seeds from a plant were investigated in the assays, and the sprouting data of every line were obtained from at least three plants. The PHS investigations were finished in 1 day.

**RT-qPCR**. The total RNA was extracted using the TRIzol™ Reagent (Invitrogen, Cat. no. 15596018). Reverse transcription was performed using the SuperScript® III Reverse Transcriptase (Invitrogen, Cat. no. 18080-044). Real-time PCR analyses were performed using the Bio-Rad CFX96 real-time PCR instrument and EvaGreen (Biotium, Cat. no. 31000). The PCR was conducted with gene-specific primers for the target genes, and Ubi-F and Ubi-R for *UBIQUITIN* (Supplementary Table 7).

**Transcriptome analyses**. Two-week-old seedlings were used in the transcriptome analyses of seedling shoots. The materials of unelongated culms, seedling shoots and seed embryos were sampled with three biological repeats for the RNA-sequencing (RNA-seq) analyses. RNAs were extracted with RNAprep pure Plant kit (TIANGEN, Cat. no. DP432), and then libraries were constructed using TruSeq Stranded mRNA (Illumina, San Diego, CA, USA) in accordance with the manufacturer's instructions. Qualities of RNA-seq libraries were assessed by using a Fragment Analyzer (Advanced Analytical, IA, USA), and the resulting libraries were sequenced using Illumina Hiseq X ten. The raw reads were filtered by removing reads containing adapter and low quality reads for subsequent analyses. Clean reads were aligned to the rice reference genome (TIGR release 7) using Hisat2 with default parameters, and resultant files were input to the Cufflinks software for comparative assembly of transcripts and generation of fragments per kilobase of exon per million reads mapped (FPKM). Subsequently, gene expression analyses between the wild type and *mir156* mutants were executed using the cufflinks-cuffdiff analysis module.

**Phytohormone measurement**. Plant materials were ground into powder in liquid nitrogen, and extracted with methanol/water (8/2) at 4 °C. The extract was centrifuged at 12,000 g under 4 °C for 15 min. The supernatant was collected and evaporated to dryness under nitrogen gas stream, and then reconstituted in methanol/water (3/7). The solution was centrifuged and the supernatant was collected for LC-MS analysis. The LC-MS analysis was conducted with the API6500 Q TRAP LC/MS/MS system, equipped with an ESI Turbo Ion-Spray interface, operating in a positive ion mode and controlled by Analyst 1.6 software (AB Sciex).

**Protein extraction and western blot**. For measuring SLR1 abundance, total proteins were extracted using a protein extraction buffer (50 mM Tris-HCl, pH 7.4,

150 mM NaCl, 2% SDS, 1 mM EDTA, 1 mM dithiothreitol, and 1 μM PMSF), and then separated in a 10% SDS-PAGE gel. After the protein sample was transferred from the SDS-PAGE gel to a HATF (Hybridization Nitrocellulose Filter) membrane (Merck Millipore, Ireland), immunodetection of SLR1 was performed with a rabbit anti-SLR1 primary antibody and an anti-IgG-HRP secondary antibody (Abmart, China). The anti-SLR1 primary antibody was kindly provided by Donglei Yang[17]. HRP signal was detected using the SuperSignal™ West Pice PLUS kit (Thermo Scientific, USA). Images were captured using ChemiDoc™ XRS + with Image Lab™ software (BIO-RAD).

**ChIP-qPCR.** A $P_{35S}:IPA1m:3 \times FLAG$ vector was constructed in the plasmid pCAMBIA1305 backbone and transformed into XS134 calli. Five $P_{35S}:IPA1m:3 \times FLAG$ seedlings of T0 generation (about 20-day-old) were used in these assays, and the seedling shoots of these five $P_{35S}:IPA1m:3 \times FLAG$ plants were pooled together to conduct the ChIP-qPCR assays. The ChIP assays were conducted based on a previous report[43]. In brief, 4 g of seedling shoots were collected from $P_{35S}:IPA1m:3 \times FLAG$ and $P_{35S}:3 \times FLAG$ seedlings grown in greenhouse (26 °C, 80% humidity, and 12 h light/12 h dark). The chromatin complexes were isolated, sonicated, and then incubated with monoclonal anti-FLAG antibodies (SIGMA). Precipitated chromatin DNA was analyzed using quantitative PCR (AceQ qPCR SYBR Green Master Mix, Vazyme). The primers used in this experiment are listed in Supplementary Table 8.

**EMSA.** The primers used to amplify GTAC-containing fragments are listed in Supplementary Table 9. GST-tagged IPA1 was purified from *Escherichia coli* BL21 (DE3) cells using Glutathione Sepharose™ 4B columns (GE Healthcare, Cat. no. 17-0756-01). Binding of IPA1 to the promoter regions of *KAO, CPS1, GNP1, SD1, SLR1, GA2ox6,* and *EUI1* was examined using 40 ng of Cy5-labeled GTAC-containing fragments mixed with 2 mg of the purified protein in 20 mM Tris (pH 7.9), 5% (v/v) glycerol, 200 mM $MgCl_2$, 0.1 M DTT, 4% (w/v) BSA, and 0.5% (w/v) salmon sperm DNA. In the competition tests, the mixture was supplemented with increasing amounts (1:10 to 1:20 or 1:10 to 1:80 mass ratio) of unlabeled DNA fragment. Native polyacrylamide gels (6% acrylamide and bis-acrylamide, 39:1) were used for electrophoresis, and the fluorescence signals were detected with a Starion FLA-9000 (FujiFilm, Japan).

**Accession numbers.** The sequence data of the *MIR156s* can be found in the mirbase database (http://www.mirbase.org/) under the following accession numbers: *MIR156a,* MI0000653; *MIR156b,* MI0000654; *MIR156c,* MI0000655; *MIR156d,* MI0000656; *MIR156e,* MI0000657; *MIR156f,* MI0000658; *MIR156g,* MI0000659; *MIR156i,* MI0000661; *MIR156h,* MI0000662; *MIR156k,* MI0001090; and *MIR156l,* MI0001091. The sequence data of the other genes can be found in the MSU database (http://rice.plantbiology.msu.edu/) under the following gene locus identifiers: *IPA1/SPL14,* LOC_Os08g39890; *SPL2,* LOC_Os01g69830; *SPL3,* LOC_Os02g04680; *SPL4,* LOC_Os02g07780; *SPL7,* LOC_Os04g46580; *SPL11,* LOC_Os06g45310; *SPL12,* LOC_Os06g49010; *SPL13,* LOC_Os07g32170; *SPL16,* LOC_Os08g41940; *SPL17,* LOC_Os09g31438; *SPL18,* LOC_Os09g32944; *GNP1,* LOC_Os03g63970; *SD1,* LOC_Os01g66100; *GA2ox4,* LOC_Os05g34854; *CPS1,* LOC_Os02g17780; *KO2,* LOC_Os06g37364; *KAO,* LOC_Os06g02019; *SLR1,* LOC_Os03g49990; *SPY,* LOC_Os08g44510; *GA2ox6,* LOC_Os04g44150; *GA2ox8,* LOC_Os05g48700; *GA2ox10,* LOC_Os05g11810; *EUI1,* LOC_Os05g40384.

**Reporting summary.** Further information on research design is available in the Nature Research Reporting Summary linked to this article.

## Data availability

The source data underlying all reported averages in chart and tables, as well as uncropped versions of blots, are provided as a Source Data file. The RNA-seq data generated in this study were deposited in the NCBI's Gene Expression Omnibus (GEO) Database under the accession number GSE131243, and the results of the RNA-seq analyses are available in Supplementary Data 2–7.

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

## Acknowledgements

This work was supported by the National Natural Science Foundation of China for Young Scientists (grant. no. 31800241), the Shanghai Center for Plant Stress Biology, Chinese Academy of Sciences, and Zhejiang A&F University. We thank Life Science Editors for editorial assistance and thank Donglei Yang for providing the SLR1 primary antibody.

## Author contributions

C.M., Z.W., and J.-K.Z. conceived and designed the research. C.M. constructed the *mir156* mutants, performed the phytohormone measurements, and conducted the phenotypic and transcriptome analyses. Z.W. conducted the Northern blotting, qRT-PCR, ChIP-qPCR, EMSA, and western blotting assays. L.Z. provided the $P_{35S}$:*IPA1* lines. C.M. and Z.W. did other assays together. K.H., X.L., and J.Y. provided assistance in this research. C.M., Z.W., and J.-K.Z. analyzed the data and wrote the paper together. H.S. helped revise the paper and provided useful suggestions. J.-K.Z. oversaw the entire study.

## Additional information

**Competing interests:** The authors declare no competing interests.

