## [Peer Review File · Nature Communications]

Reviewers' comments:

Reviewer #1 (Remarks to the Author):

I think this work is interesting and exploits careful disruption of miR156 precursor genes to understand regulation of seed yield traits in rice. The authors find that loss of miR156 results in the expected upregulation of IPA1, and that overexpression of IPA1 phenocopies the miR156 loss of function lines. Reductions in pre-harvest sprouting and increased dormancy are also shown, and data is presented that suggests that this is through GA regulation.

In general I found the data supported the conclusions of the authors. However, there is one area where I was unclear.

The mis-regulation of the GA pathway in miR156 disruption lines and IPA1 overexpressors appeared to involve different genes than the gene which IPA1 bound by ChIP. Of course the authors have focussed on known IPA1 binding site sequences, but there may well be others depending on binding partner specificity. So it remains important to rule the genes out where miR156 and IPA1 cause large mis-regulation in lines with the dormancy phenotype. ChIP seq could be the obvious approach.

Reviewer #2 (Remarks to the Author):

MS 198316

TITLE: miR156, a plant architecture and grain size modulator, regulates seed dormancy through the gibberellin pathway in rice

This manuscript describes the characterization of rice mutant lines generated by CRISPR-Cas9 genome editing aimed at knocking out eleven MIR156 genes present in two japonica cultivars. The results obtained indicate that this large MIRNA gene family can be divided into two sub-groups based on sequence relatedness and ultimately in phenotypic relationships found when multiple gene knockouts were analyzed. Group I gene mutants (MIR156d-i) resulted in plants with changes in shoot

architecture (taller and stronger plants with fewer tillers) while group II mutants (MIR156abckl) were unaffected in these traits.

Among other phenotypes, group II mutants and MIR156a-l (except e) showed slower germination. Consistently, group II mutants showed reduced post-harvesting sprouting (PHS). Also, seeds from group II mutants showed increased seed viability with time. All these results indicate that group II mutants have extended seed dormancy.

However, ABA levels in mutants were lower than in WT although crossing MIR156 mutants with PYL mutants did not result in enhanced germination in PYL/MIR156 mutant lines, suggesting that the effect of MIR156 mutants is not through ABA signaling.

RNA-seq of WT and MIR156a-l (except e) mutant fresh embryos showed that among differentially expressed genes many GA-related genes were altered in the mutant. IPA1/OsSPL14 (a transcription factor) is targeted by miR156, thus in RNA-seq data it was found with increased accumulation in MIR156 mutant fresh embryos.

To explore the genes targeted for regulation by IPA1, the p35S:flag:IPA1 overexpressing plants were used to carry out ChIP-qPCR, which showed binding of IPA1 to promoters of GA-related DEGs, making a case for miR156 regulation of GA-related genes through the regulation of IPA1.

The authors carried out a systematic analysis of all MIR156 loci in rice through CRISPR-Cas9 genome editing. It represents a real tour de force as the multiple members of a microRNA gene family make it difficult to uncover MIRNA functions due to potential redundancy. The identification of the IPA1 gene as the main responsible factor of multiple effects observed in the mutant lines together with the identification of targeted GA-related genes represents an advancement of the field.

The knowledge is relevant for miRNA field as it uncovers subtle effects of individual gene members of the MIR156 family and it provides tools to study PHS that could be transferred to rice breeders and likely applicable to other crops as well. Previous works had already explored the roles of IPA1 in plant architecture and its regulation by miR156 (Miura et al, 2010; Jiao et al, 2010), but the unbiased approach to define the effect of miR156 regulation had not been explored in detail.

In addition, there are a number of points that should be addressed by the authors to have a more comprehensive story.

1. Group II mutants (mir156 a-l) showed reduced levels of ABA. Seed dormancy and PHS have been extensively linked to ABA metabolism and perception (see for example Bewley JD: Seed germination and dormancy. Plant Cell 1997; Koornneef M, Bentsink L, Hilhorst H: Seed dormancy and germination. Curr Opin Plant Biol 2002, Gubler F, Millar AA, Jacobsen JV: Dormancy release, ABA and pre-harvest sprouting. Curr Opin Plant Biol 2005). It results counterintuitive that ABA levels are reduced in mir156 mutants and yet they show extended seed dormancy.

To try to dissect this effect, the authors used multiple PYL mutants previously published (Miao et al 2018, PNAS). However, according to that report the pyl mutant selected (pyl 1/4/6) was among those that showed little effect in PHS compared to WT plants, thus it is unlikely to have a clear phenotype when compared to the mir156 mutants in the crossed plants shown in the present manuscript. Perhaps the authors should have used the pyl12 mutant which was clearly affected as shown before (Miao et al, 2018).

Independently of this fact, the reduced ABA levels in mir156 mutants is not explained in the manuscript. Thus, a number of important questions are unresolved: Are there other defects in the mutant lines due to this alteration in ABA such as impaired stress responses? If this is the case, then the argument towards use of mir156 mutants or even IPA1 for seed improvement should be reconsidered. Are there any plant fitness penalties as a consequence of altered ABA levels and IPA1 deregulated accumulation?

2. As expected, miR156 mutants cause SPL genes to become deregulated. Significantly, IPA1 is the most affected, consistent with previous results in rice where IPA1 was characterized as target of mir156 regulation (Jiao et al 2010). In that report a mutant allele of IPA1 with point mutation rendering IPA1 insensitive to miR156 binding showed its effects in plant architecture. Further, the point mutation in OsSPL14 perturbs OsmiR156-directed regulation of OsSPL14, generating an 'ideal' rice plant with a reduced tiller number, increased lodging resistance and enhanced grain yield (Jiao et al 2010, Nat Genet). Thus, identification of IPA1 as target of miR156 is not novel (see also Miura et al, 2010 with similar results).

In contrast, in the present manuscript the overexpression of IPA1 using the pOle1:IPA1 construct did not show plant architecture defects (contrary to what was published using the Shaonieijing variety in Jiao et al 2010). While plants carrying the p35S:IPA1 construct had many defects. Thus, the claim that IPA1 can be used as a tool for seed improvement seems for the moment to depend on genetic background, a consideration that should be at least mentioned in the present text.

In addition, to fully appreciate the different transgenic plants employed and their significance, it would be required to compare levels of IPA1 achieved by over-expression (pOle1 or p35S) and those achieved in the miR156 mutants. This piece of data would then allow the reader to more accurately compare the results offered in the manuscript and to integrate the results shown.

3. Another concern is the use of the p35S:flag:IPA1 construct in ChIP assays. It has the disadvantage that the over-expression levels attained might be above the endogenous levels and thus cause binding of IPA1 to sequence elements that wt IPA1 would not do under normal conditions (gain of function). As stated in my previous point, one would need to determine the IPA1 levels achieved by miR156 mutation and those achieved by IPA1 overexpression to more accurately understand the effects observed in GA-related gene expression and phenotypic features.

A more real result would be achieved by having a tagged IPA1 expressed from the native promoter.

Minor points

1. Throughout the manuscript the word 'homologous' is used when 'homozygous' should be the correct one.

2. While the identity of the multiple mir156 mutants is essential for the interpretation of the results, little is mentioned as to how the mutants were genotypically characterized. Supplementary information describes the strategy to generate the mutants through genome editing, but the assays to determine molecular lesions (and absence of unwanted off-targets) are not described or referenced.

3. In the current version of the manuscript it is hard to follow which genotype (Nipponbare or XS134) was used for individual experiments, and towards the last experiments XS134 is not mentioned anymore.

This is especially important as there are potential differences between accessions if phenotypes are being scored. For instance, it was previously shown that altered expression of IPA1 has phenotypic consequences in different backgrounds: Nipponbare vs. ST-12 (Miura et al, 2010), Shaoniejing variety (Jiao et al 2010) and results shown here with Nipponbare and XS134.

Reviewer #3 (Remarks to the Author):

miR156, a plant architecture and grain size modulator, regulates seed dormancy through the gibberellin pathway in rice

Reviewer:

Now, miR156/SPLs module which is well-conserved in high plant utilization in breeding is exemplified in rice production. Although some of SPLs genes function in rice are revealed by QTL genetic analysis and functional dissection such as IPA1, SPL16, SPL13, the upstream miR156 specific function is unknown. Overexpression of miR156 generally lead to the same high-tillers and late heading phenotype in high plant. Is there also exist distinct function of miR156s similar with its

target gene-SPLs? How to discern the different function of miR156s? I think this manuscript gave a good example to answer these questions.

In the manuscript, the authors utilize powerful Crispr-Cas gene-editing tool to make a series of mir156 mutant combination. Phenotyping and function analysis of these combinational mutant, they succeed in classifying two group of miR156 which are responsible for distinct function. The experimental data are profound and trustable. To clearly describe my evaluation of this manuscript, I will list my remarks as below as followed:

- (1) A series of mir156s knock-out mutants you have gotten, why not these mutational sites clearly showed at the targeted localization? Please supplement these mutational sites in all knock-out mir156s mutants
- (2) According to your results, you mentioned that GroupI mir156s lead to large grain size in figure 2.a, however, I could not discern the seed size difference in this figure. Should you offer an obvious grain size comparison photo among these lines? Meanwhile, you also mentioned that mirfg is a valuable genetic resource for grain yield improvement which is an important for future breeding! Did you check which SPL(s) gene(s) expression and miR156 expression change in vivo also?
- (3) In supplementary fig2, MIR156I repeat two times, please corrected it. I know miR156h/I is transcript from the same gene, but annotation sequences are different between the two primary transcripts.
- (4) Did you check leaf emergence rate of mirabckl and WT? Based on phenotypic observation at juvenile stage of mirabckl, I suggest you check which SPLs gene expression change which may helpful for discern different function of SPLs genes.
- (5) In figure1I, MIR156 expression level appear to be no change between WT and mirabckl, but sharply decrease in mirdefghi mutant at 25-day-old seedlings. In figure3c, you showed mirabckl is very low germination rate, but mirdefghi mutant has some nearly normal germination rate. Here, I strong suggest you recheck miR156 expression level among mirabckl, mirdefghi and WT at germination stage. In addition, SPLs gene expression are better to be checked at this stage.
- (6) In rice, SLR1 antibody is available, it is better to confirm SLR1 protein accumulation in mirabckl mutant which offer further evidence to support your conclusion that mir156 mutations enhanced seed dormancy by suppressing GA pathway.
- (7) Now, epi-allele ipa1-2 germplasm with high SPL expression has widely utilization in Chinese hybrid breeding. In this manuscript further revealed new function SPL14 gene related to resistance to PHS. I think in your discussion part, you may draw a specific utilization strategy how to use mirfg and miR156/SPLs module in future rice breeding and other crop plants.

In a word, after the manuscript complete some revision, I think it is suitable for publish on Nature Communication.

Response to Reviewers' Comments

Dear editors and reviewers,

Thank you very much for your comments and suggestions on our manuscript (NCOMMS-19-04295-T) entitled “**miR156, a plant architecture and grain size modulator, regulates seed dormancy through the gibberellin pathway in rice**”. The manuscript has been revised according to the editorial policies and reviewers' suggestions, and the revisions in the manuscript are highlighted in red. Below please find our point-by-point responses to the reviewers' comments.

Reviewer #1 (Remarks to the Author):

I think this work is interesting and exploits careful disruption of miR156 precursor genes to understand regulation of seed yield traits in rice. The authors find that loss of miR156 results in the expected upregulation of IPA1, and that overexpression of IPA1 phenocopies the miR156 loss of function lines. Reductions in pre-harvest sprouting and increased dormancy are also shown, and data is presented that suggests that this is through GA regulation.

In general I found the data supported the conclusions of the authors. However, there is one area where I was unclear.

The mis-regulation of the GA pathway in miR156 disruption lines and IPA1 overexpressors appeared to involve different genes than the gene which IPA1 bound by ChIP. Of course the authors have focussed on known IPA1 binding site sequences, but there may well be others depending on binding partner specificity. So it remains important to rule the genes out where miR156 and IPA1 cause large mis-regulation in lines with the dormancy phenotype. ChIP seq could be the obvious approach.

Response: We thank reviewer for the comments and suggestions.

We agree that mis-regulation of the GA pathway in *MIR156* disruption lines and *IPA1* overexpression lines might involve genes which are not bound by IPA1. In fact, although we detected *in vivo* associations of IPA1 with the promoters of many GA-related DEGs (differentially expressed genes), we did not detect interactions between IPA1 and the promoters of two GA-related DEGs (DEGs between wild-type and *mir156abcdghijkl* seedling shoots), *GA20ox4*

and *GA2ox10*. We propose that these two genes (*GA2ox4* and *GA2ox10*) may be bound and regulated directly by SPL proteins encoded by other miR156 target genes, or regulated indirectly by IPA1. We have added these points to our revised manuscript (see the last part of the “Results” section). In addition, IPA1 was recently reported to directly interact with SLR1, a negative regulator in GA pathway (Liu *et al.*, 2019, *Nature Plants* 5, 389-400). Thus, miR156 could also affect the GA pathway through direct interaction between IPA1 and SLR1. Furthermore, we also agree that IPA1 may also bind and thus directly regulate the GA-related genes through unknown IPA1-binding DNA elements.

The data in our manuscript show that miR156 regulates seed dormancy and seedling growth through a molecular network composed of *IPA1* and many downstream genes in the GA pathway. Considering the similarity between IPA1 and other miR156-targeting SPLs, as well as mis-regulations of these *SPL* genes in *mir156* mutants (**Fig. 6a** and **Supplementary Table 5**), it is plausible that miR156 also regulates the GA pathway through other miR156 target genes (*SPL2–SPL4*, *SPL7*, *SPL11–SPL13*, and *SPL16–SPL18*). Thus, our data should prompt the readers of our manuscript to speculate that miR156 regulates seed dormancy and plant growth through a complex molecular network which composes of *SPLs* (including *IPA1*) and many downstream genes in the GA pathway. We have added these notions to our revised manuscript (see the fifth paragraph of the “Discussion” section).

IPA1 ChIP-seq has been conducted in detail by Jiayang Li’s group (Lu *et al.*, 2013, *Plant Cell* 25, 3743-3759). We have cited this work in our manuscript (reference 33). The ChIP-seq data by Jiayang Li’s team suggest that IPA1 binds and directly regulates some GA-related genes (including *EUI1*, *GA2ox6*, *SLR1*, *GSRI*, *GID2*, and two putative GA receptor genes LOC_Os02g35940 and LOC_Os06g20200). All of these IPA1-binding genes contain the GTAC or TGGGCC/T motifs (IPA1-binding motifs) in their 1000-bp promoter regions.

Reviewer #2 (Remarks to the Author):

MS 198316

TITLE: miR156, a plant architecture and grain size modulator, regulates seed dormancy through the gibberellin pathway in rice

This manuscript describes the characterization of rice mutant lines generated by CRISPR-Cas9 genome editing aimed at knocking out eleven *MIR156* genes present in two *japonica* cultivars. The results obtained indicate that this large *MIRNA* gene family can be divided into two sub-groups based on sequence relatedness and ultimately in phenotypic relationships found when

multiple gene knockouts were analyzed. Group I gene mutants (*MIR156d-i*) resulted in plants with changes in shoot architecture (taller and stronger plants with fewer tillers) while group II mutants (*MIR156abckl*) were unaffected in these traits.

Among other phenotypes, group II mutants and *MIR156a-l* (except e) showed slower germination. Consistently, group II mutants showed reduced post-harvesting sprouting (PHS). Also, seeds from group II mutants showed increased seed viability with time. All these results indicate that group II mutants have extended seed dormancy.

However, ABA levels in mutants were lower than in WT although crossing *MIR156* mutants with *PYL* mutants did not result in enhanced germination in *PYL/MIR156* mutant lines, suggesting that the effect of *MIR156* mutants is not through ABA signaling.

RNA-seq of WT and *MIR156a-l* (except e) mutant fresh embryos showed that among differentially expressed genes many GA-related genes were altered in the mutant. *IPA1/OsSPL14* (a transcription factor) is targeted by miR156, thus in RNA-seq data it was found with increased accumulation in *MIR156* mutant fresh embryos.

To explore the genes targeted for regulation by IPA1, the *P_{35S}:flag:IPA1* overexpressing plants were used to carry out ChIP-qPCR, which showed binding of IPA1 to promoters of GA-related DEGs, making a case for miR156 regulation of GA-related genes through the regulation of IPA1.

The authors carried out a systematic analysis of all *MIR156* loci in rice through CRISPR-Cas9 genome editing. It represents a real tour de force as the multiple members of a microRNA gene family make it difficult to uncover *MIRNA* functions due to potential redundancy. The identification of the *IPA1* gene as the main responsible factor of multiple effects observed in the mutant lines together with the identification of targeted GA-related genes represents an advancement of the field.

The knowledge is relevant for miRNA field as it uncovers subtle effects of individual gene members of the *MIR156* family and it provides tools to study PHS that could be transferred to rice breeders and likely applicable to other crops as well. Previous works had already explored the roles of *IPA1* in plant architecture and its regulation by miR156 (Miura et al, 2010; Jiao et al, 2010), but the unbiased approach to define the effect of miR156 regulation had not been explored in detail.

In addition, there are a number of points that should be addressed by the authors to have a more comprehensive story.

1. Group II mutants (*mir156 a-l*) showed reduced levels of ABA. Seed dormancy and PHS have been extensively linked to ABA metabolism and perception (see for example Bewley JD: Seed

germination and dormancy. Plant Cell 1997; Koornneef M, Bentsink L, Hilhorst H: Seed dormancy and germination. Curr Opin Plant Biol 2002, Gubler F, Millar AA, Jacobsen JV: Dormancy release, ABA and pre-harvest sprouting. Curr Opin Plant Biol 2005). It results counterintuitive that ABA levels are reduced in *mir156* mutants and yet they show extended seed dormancy.

To try to dissect this effect, the authors used multiple *PYL* mutants previously published (Miao et al 2018, PNAS). However, according to that report the *pyl* mutant selected (*pyl1/4/6*) was among those that showed little effect in PHS compared to WT plants, thus it is unlikely to have a clear phenotype when compared to the *mir156* mutants in the crossed plants shown in the present manuscript. Perhaps the authors should have used the *pyl12* mutant which was clearly affected as shown before (Miao et al, 2018).

Independently of this fact, the reduced ABA levels in *mir156* mutants is not explained in the manuscript. Thus, a number of important questions are unresolved: Are there other defects in the mutant lines due to this alteration in ABA such as impaired stress responses? If this is the case, then the argument towards use of *mir156* mutants or even *IPAI* for seed improvement should be reconsidered. Are there any plant fitness penalties as a consequence of altered ABA levels and *IPAI* deregulated accumulation?

Response: We thank reviewer for the critical comments and questions.

Regarding the *pyl1/4/6* mutant, Miao et al. examined pre-harvest sprouting (PHS) just after seed-filling stage, and at this stage, they found very slightly higher PHS frequencies in *pyl1/4/6* than in the wild type (Miao *et al*, 2018, PNAS 115, 6058-6063). We also did not observe substantial differences in PHS between *pyl1/4/6* and the wild type just after the seed-filling stage.

However, we found that at the normal harvest time in Hangzhou, PHS frequencies were obviously higher in *pyl1/4/6* than in the wild type (about 21% in *pyl1/4/6* vs 12% in the wild type). In addition, at the sowing time in the paddy field, seed germination was obviously accelerated for *pyl1/4/6* compared to the wild type, further confirming impaired seed dormancy in *pyl1/4/6*.

Thus, although *pyl1/4/6* mutations can promote rice growth and increase productivity, the defect in seed dormancy poses a potential problem for its application (Miao *et al*, 2018, PNAS 115, 6058-6063). Here, we found that *mir156abckl* mutations inhibit the seed dormancy defect in *pyl1/4/6* without obviously affecting other traits. This finding should promote the application of *pyl1/4/6* mutations in agriculture.

ABA is an important stress phytohormone that enhances plant adaptation to abiotic and biotic

stresses. In *mir156abcdghijkl* and *mir156abckl*, we observed decreased ABA level in seeds, indicating that miR156 may positively regulate ABA level. However, in *mir156* mutants, stress resistance does not appear to be compromised. Deficiency in GA pathway can improve abiotic stress resistance (Colebrook *et al*, 2013, *J.Exp.Biol.* 217, 67-75). In *mir156* mutants, impaired GA pathway was observed. Consistent with the impaired GA pathway, we found that group I *mir156* mutations reduced transpirational water loss rates of detached flag leaf blades and improved drought stress tolerance (See **Supplementary Figs. 22 and 23**). In addition, it was recently reported that miR156 knockdown or *IPA1* overexpression conferred resistance against rice blast and bacterial blight, two major devastating diseases in rice (Wang *et al*, 2018, *Science* **361**, 1026-1028; Liu *et al*, 2019, *Nature Plants* 5, 389-400). Thus, *MIR156* knockout through gene editing may improve not only the agronomic traits but also stress resistance. We have added these points to the “Discussion” section of our revised manuscript (see the last paragraph of the “Discussion” section).

2. As expected, miR156 mutants cause *SPL* genes to become deregulated. Significantly, *IPA1* is the most affected, consistent with previous results in rice where *IPA1* was characterized as target of miR156 regulation (Jiao *et al* 2010). In that report a mutant allele of *IPA1* with point mutation rendering *IPA1* insensitive to miR156 binding showed its effects in plant architecture. Further, the point mutation in OsSPL14 perturbs OsmiR156-directed regulation of OsSPL14, generating an 'ideal' rice plant with a reduced tiller number, increased lodging resistance and enhanced grain yield (Jiao *et al* 2010, *Nat Genet*). Thus, identification of *IPA1* as target of miR156 is not novel (see also Miura *et al*, 2010 with similar results).

In contrast, in the present manuscript the overexpression of *IPA1* using the pOle1:*IPA1* construct did not show plant architecture defects (contrary to what was published using the Shaonieijing variety in Jiao *et al* 2010). While plants carrying the p35S:*IPA1* construct had many defects. Thus, the claim that *IPA1* can be used as a tool for seed improvement seems for the moment to depend on genetic background, a consideration that should be at least mentioned in the present text.

In addition, to fully appreciate the different transgenic plants employed and their significance, it would be required to compare levels of *IPA1* achieved by over-expression (pOle1 or p35S) and those achieved in the miR156 mutants. This piece of data would then allow the reader to more accurately compare the results offered in the manuscript and to integrate the results shown.

Response: We thank reviewer for the comments and suggestions.

Ole18 promoter is a seed-specific promoter (see reference 32), and consistently, *IPAI* overexpression using the *P_{Ole18}:IPAI* construct did not lead to obvious changes in plant architecture. Thus, we think that the lack of effects of *P_{Ole18}:IPAI* on plant architecture is due to the tissue specificity of *Ole18* promoter rather than the genetic background.

We used RT-qPCR to compare *IPAI* expression between *mir156abcdghikl* and *IPAI* overexpression lines (*P_{35S}:IPAI*, *P_{35S}:IPAI_m:3×FLAG*, or *P_{Ole18}:IPAI*). The results have been added to the revised manuscript (See **Fig. 6e**, **Supplementary Fig.19b**, and **Supplementary Fig. 20a**).

3. Another concern is the use of the p35S:flag:IPAI construct in ChIP assays. It has the disadvantage that the over-expression levels attained might be above the endogenous levels and thus cause binding of IPAI to sequence elements that wt IPAI would not do under normal conditions (gain of function). As stated in my previous point, one would need to determine the *IPAI* levels achieved by miR156 mutation and those achieved by *IPAI* over-expression to more accurately understand the effects observed in GA-related gene expression and phenotypic features. A more real result would be achieved by having a tagged IPAI expressed from the native promoter.

Response: We thank reviewer for this excellent question.

We have compared *IPAI* expression in the seedling shoots of *mir156abcdghikl* and *P_{35S}:IPAI_m:3×FLAG*, and found that *IPAI* expression level was substantially increased by both *mir156abcdghikl* mutations and *P_{35S}:IPAI_m:3×FLAG* overexpression (see **Supplementary Fig. 20a**). *IPAI* expression level was increased even more intensely in *mir156abcdghikl* at early seedling stage (four-day-old) than in the *P_{35S}:IPAI_m:3×FLAG* over-expression seedlings used for the ChIP-qPCR assays (see **Supplementary Fig. 20a**). Thus, we think that the ChIP-qPCR results should basically reflect the native situation in *mir156* mutants.

Minor points

1. Throughout the manuscript the word ‘homologous’ is used when ‘homozygous’ should be the correct one.

Response: The word “homologous” has been replaced by “homozygous”. We thank reviewer for pointing this out.

2. While the identity of the multiple mir156 mutants is essential for the interpretation of the results, little is mentioned as to how the mutants were genotypically characterized. Supplementary information describes the strategy to generate the mutants through genome editing, but the assays to determine molecular lesions (and absence of unwanted off-targets) are not described or referenced.

Response: We thank reviewer for this comment.

We have added the method for genotypic characterization to the “Plant material construction and cultivation” part in “**Methods**” section.

We designed sgRNAs specific for the *MIR156* genes to avoid unwanted off-target mutations in other genes. We also used two or more independent mutant lines for phenotyping, which should minimize the effects of possible off-target mutations on the mutant phenotypic analyses. Considering that the sgRNAs designed for certain *MIR156* genes may induce off-target mutations in other *MIR156* members (for example, the sgRNA for *MIR156a* also induces mutations in *MIR156b* and *MIR156c*. See **Supplementary Fig. 1a–c**), we sequenced both the target sites and potential off-target sites in *MIR156* genes to accurately identify the genotype. These points have been added to the “Plant material construction and cultivation” part in “**Methods**” section.

3. In the current version of the manuscript it is hard to follow which genotype (Nipponbare or XS134) was used for individual experiments, and towards the last experiments XS134 is not mentioned anymore. This is especially important as there are potential differences between accessions if phenotypes are being scored. For instance, it was previously shown that altered expression of *IPA1* has phenotypic consequences in different backgrounds: Nipponbare vs. ST-12 (Miura et al, 2010), Shaoniejing variety (Jiao et al 2010) and results shown here with Nipponbare and XS134.

Response: We thank reviewer for the comments. In all figures, tables and datasets (data in Excel files), the backgrounds (Nipponbare, XS134, or ZH11) of all the assays have been clearly indicated (see the controls of the assays).

Reviewer #3 (Remarks to the Author):

miR156, a plant architecture and grain size modulator, regulates seed dormancy through the gibberellin pathway in rice

Reviewer:

Now, miR156/SPLs module which is well-conserved in high plant utilization in breeding is exemplified in rice production. Although some of *SPLs* genes function in rice are revealed by QTL genetic analysis and functional dissection such as *IPA1*, *SPL16*, *SPL13*, the upstream miR156 specific function is unknown. Overexpression of miR156 generally lead to the same high-tillers and late heading phenotype in high plant. Is there also exist distinct function of miR156s similar with its target gene-*SPLs*? How to discern the different function of miR156s? I think this manuscript gave a good example to answer these questions.

In the manuscript, the authors utilize powerful Crispr-Cas gene-editing tool to make a series of mir156 mutant combination. Phenotyping and function analysis of these combinational mutant, they succeed in classifying two group of miR156 which are responsible for distinct function. The experimental data are profound and trustable. To clearly describe my evaluation of this manuscript, I will list my remarks as below as followed:

(1) A series of *mir156s* knock-out mutants you have gotten, why not these mutational sites clearly showed at the targeted localization? Please supplement these mutational sites in all knock-out *mir156s* mutants

Response: We thank reviewer for the suggestion. The mutated sequences of every mutant are shown in **Supplementary Data 1a and 1b**. All the mutations occurred in the target sites.

(2) According to your results, you mentioned that GroupI *mir156s* lead to large grain size in figure 2.a, however, I could not discern the seed size difference in this figure. Should you offer an obvious grain size comparison photo among these lines? Meanwhile, you also mentioned that *mirfg* is a valuable genetic resource for grain yield improvement which is an important for future breeding! Did you check which *SPL(s)* gene(s) expression and miR156 expression change in vivo also?

Response: We thank reviewer for the questions.

We have replaced original **Fig. 2a** with a new photo. The original **Fig. 2a** and the new **Fig. 2a** are present here side-by-side for comparison.

miR156fg showed similar plant shoot architecture with the wild type. Consistently, miR156 abundance and the expression of all miR156 target genes do not show obvious differences between wild-type and *mir156fg* seedling shoots (see **Supplementary Fig. 5a–I**).

(3) In supplementary fig2, *MIR156I* repeat two times, please corrected it. I know miR156h/I is transcript from the same gene, but annotation sequences are different between the two primary transcripts.

Response: We are very sorry for this mistake. The mistake has been corrected (see **Supplementary Fig. 2**).

(4) Did you check leaf emergence rate of *mirabckl* and WT? Based on phenotypic observation at juvenile stage of *mirabckl*, I suggest you check which *SPLs* gene expression change which may helpful for discern different function of *SPLs* genes.

Response: We thank reviewer for the question and suggestion.

MiR156 positively regulates leaf emergence rate in rice. Since this information has been shown in several previous reports, we did not spend much time on investigating the leaf emergence rates of the *mir156* mutants. We indeed noticed that group I *mir156* mutations obviously reduced leaf emergence rate. Because we did not observe substantial differences in plant shoot architecture between the wild type and *mir156abckl*, we did not check leaf emergence rate of *mir156abckl* vs the wild type.

We have checked the expression of miR156 target genes (11 *SPL* genes) in *mir156abckl* and *mir156defghi* at the juvenile stage (20-day-old seedlings). Tillers first grow from the unelongated culms in seedlings. Therefore, we checked the expression of miR156 target genes in the unelongated culms of 20-day-old seedlings via transcriptome analyses. The transcriptome analyses showed that the expression of the miR156 target genes was not highly up-regulated by *mir156abckl* mutations (ratio < 2), whereas expression of several miR156 target genes (*SPL3*, *SPL13*, *IPA1/SPL14*, and *SPL17*) was markedly up-regulated by *mir156defghi* (ratio > 2) (see **Supplementary Fig. 7a, b**). These data have been added to the revised manuscript.

(5) In figure 1I, *MIR156* expression level appear to be no change between WT and *mirabckl*, but sharply decrease in *mirdefghi* mutant at 25-day-old seedlings. In figure 3c, you showed *mirabckl* is very low germination rate, but *mirdefghi* mutant has some nearly normal germination rate. Here, I strong suggest you recheck miR156 expression level among *mirabckl*, *mirdefghi* and WT at germination stage. In addition, *SPLs* gene expression are better to be checked at this stage.

Response: We thank reviewer for the suggestions.

We have checked miR156 expression in germinating embryos of the wild type, *miR156abckl* and *mir156abcdfghikl* through Northern blots, and found that miR156 expression was obviously decreased in *miR156abckl* and *mir156abcdfghikl* (see **Supplementary Fig. 12a**). We are very sorry that we do not have enough seeds of *mir156defghi* to conduct the Northern blot analysis, because we used most of *mir156defghi* seeds in other assays. We hope the reviewer agree that lack of this data does not reduce the novelty and significance of this research.

We have checked the expression of miR156 target genes through RT-qPCR in the seed embryos of *miR156abckl*, *mir156defghi*, and the wild type. The results have been added to the manuscript (see **Supplementary Fig. 12b–k**). The RT-qPCR results revealed that several miR156 target genes (*SPL12*, *SPL13*, and *IPA1*) were more intensely up-regulated by *mir156abckl* mutations than by *mir156defghi* mutations. There are also some miR156 target genes (*SPL3*, *SPL11*, *SPL16*, *SPL17*, and *SPL18*) that were more intensely up-regulated by *mir156defghi* mutations than by *mir156abckl* mutations. These results suggest that group I and group II *MIR156s* may mainly function in different parts of the seed embryos, thus leading to the surprising expression patterns of *SPL* genes between *mir156abckl* and *mir156defghi*.

(6) In rice, SLR1 antibody is available, it is better to confirm SLR1 protein accumulation in *mirabckl* mutant which offer further evidence to support your conclusion that *mir156* mutations enhanced seed dormancy by suppressing GA pathway.

Response: We thank reviewer for the suggestion. In fresh seed embryos, *SLR1* expression was not substantially different between the wild type and *mir156abcdfghikl* (NIP_FPKM, 25.6309; *mir156abcdfghikl*_FPKM, 21.2868) (transcriptome result from **Supplementary Data 2**). Therefore, we did not perform Western blots to determine SLR1 accumulation in wild-type, *mir156abckl*, *mir156defghi* and *mir156abcdfghikl* seed embryos. However, in seedling shoots, *SLR1* expression was slightly increased in *mir156abcdfghikl* compared with the wild type

(NIP_FPKM, 97.0376; *mir156abcdghijkl*_FPKM, 182.694) (transcriptome result from **Supplementary Data 6**). Consistently, Western blotting assays suggest slightly increased SLR1 level in *mir156abcdghijkl* seedling shoots compared with the wild type (see **Supplementary Fig. 17**), further supporting that *mir156* mutations affect seedling growth through the GA pathway. SLR1 accumulation does not show obvious differences between wild-type, *mir156abckl* and *mir156defghi* seedling shoots (see **Supplementary Fig. 17**).

(7) Now, epi-allele ipa1-2 germplasm with high *SPL* expression has widely utilization in Chinese hybrid breeding. In this manuscript further revealed new function *SPL14* gene related to resistance to PHS. I think in your discussion part, you may draw a specific utilization strategy how to use *mirfg* and miR156/SPLs module in future rice breeding and other crop plants.

In a word, after the manuscript complete some revision, I think it is suitable for publish on Nature Communication.

Response: We highly appreciate the reviewer's suggestion and comment.

In the first paragraph of the "Discussion" section in our manuscript, we discussed how to use gene editing of *MIR156s* to adjust plant architecture and grain size according to the dosage breeders need and without undesirable pleiotropic effects.

In addition, in the third paragraph of the "Discussion" section, we discussed the utilization of group II *MIR156* gene editing to specifically inhibit pre-harvest sprouting (PHS) and increase seed longevity.

Furthermore, in the fourth paragraph of the "Discussion" section, we added a discussion about how to use gene editing of *MIR156s* to improve plant architecture, grain size, and seed dormancy simultaneously, and thus generate PHS-resistant rice crops with increased grain productivity. Owing to the high conservation of miR156 in plants, these strategies in rice improvement may be also applicable to other crops.

REVIEWERS' COMMENTS:

Reviewer #2 (Remarks to the Author):

The current version of the manuscript has addressed most of the comments that I originally had for the first version. It is a study relevant to different areas of Plant Biology and ready for publication.

Still, there is one aspect that I asked about which remains unresolved in my mind. In the mutant plants miR156abcdnfghijkl and miR156abckl the levels of ABA are reduced. In contrast, the authors indicate that they didn't find abnormalities related to stress responses, which helps in their argument to propose miR156 as regulator of seed dormancy and as a trait of agronomic use. Along these lines, the authors document the impact of GA defects on stress responses in the Discussion section of the revised manuscript. In addition, one of my questions was how to explain this reduced ABA level in the miR156 mutant background. I understand this is not the focus of the work, but the authors could look into the RNA-seq data and try to explain this defect. It would be nice to see some thoughts about this issue, could it be pointing towards a cross-talk between GA and ABA? Rather than seeing some speculation in the Discussion section, there might be some relevant data to be mined in the transcriptomic data.

Reviewer #3 (Remarks to the Author):

After revision, i think the authors had answered my questions.

I think different MIR156s function in rice were demonstrated by this paper.I appreciated the work

Response to reviewer comments

REVIEWERS' COMMENTS:

Reviewer #2 (Remarks to the Author):

The current version of the manuscript has addressed most of the comments that I originally had for the first version. It is a study relevant to different areas of Plant Biology and ready for publication. Still, there is one aspect that I asked about which remains unresolved in my mind. In the mutant plants *miR156abcdghijkl* and *miR156abckl* the levels of ABA are reduced. In contrast, the authors indicate that they didn't find abnormalities related to stress responses, which helps in their argument to propose miR156 as regulator of seed dormancy and as a trait of agronomic use. Along these lines, the authors document the impact of GA defects on stress responses in the Discussion section of the revised manuscript. In addition, one of my questions was how to explain this reduced ABA level in the miR156 mutant background. I understand this is not the focus of the work, but the authors could look into the RNA-seq data and try to explain this defect. It would be nice to see some thoughts about this issue, could it be pointing towards a cross-talk between GA and ABA? Rather than seeing some speculation in the Discussion section, there might be some relevant data to be mined in the transcriptomic data.

Response: We thank the reviewer for the comment and suggestion.

In *miR156abcdghijkl* and *miR156abckl* seeds, we found reduced ABA levels compared to the wild type. Whether the ABA levels in other parts of the plants are affected by *mir156* mutations is not known. In addition, the impaired GA pathway should enhance stress resistance in *mir156* mutants. Therefore, it is not very surprising that we did not find abnormalities related to stress responses in *mir156* mutants.

ABA and GAs act antagonistically on seed dormancy, and there is a sophisticated balance between the effects of ABA and GAs in seeds. In this project, we found that mutations in rice *MIR156* genes impaired the GA pathway through the miR156 target gene *IPA1*, which directly regulates many genes in the GA pathway. To reach a new balance between the effects of GAs and ABA, the *mir156* mutants may adjust the ABA pathway to antagonize the effects of the impaired GA pathway in seeds. Therefore, we speculate that the decreased ABA level in seeds of *mir156* mutants is a response to the impaired GA pathway. We have added this speculation in the last paragraph of the "Discussion" section. Of course, there may also be other reasons for the reduced ABA levels in the mutant seeds.

We also searched the transcriptomic data of wild-type and *mir156abcdghijkl* seed embryos, and found that the known ABA biosynthetic genes were not markedly down-regulated by *mir156abcdghijkl* mutations (**Table 1**). In addition, the known ABA deactivating genes (*ABA8ox1*, *ABA8ox2* and *ABA8ox3*) were not up-regulated by *mir156abcdghijkl* mutations (**Table 1**). Such, the decreased ABA levels in *mir156* seeds cannot be explained from the changed expression of the known ABA metabolic genes (biosynthetic and deactivating genes). Considering that the identification of ABA metabolic genes in rice is far from complete, unknown ABA metabolic

genes may be responsible for the decreased ABA levels in *mir156* seeds. In addition, there may be other reasons for the decreased ABA levels: such as decreased levels of ABA precursors or enhanced ABA transfer from seeds to other parts of the *mir156* mutants.

Because “how ABA levels are reduced in *mir156abcdghijkl* and *mir156abckl* seeds” is not the focus of our manuscript, we did not take many words to discuss this question in the manuscript.

Table 1 Expression profiles of known ABA metabolic genes in wild-type and *mir156abcdghijkl* fresh seed embryos

	Genes	Gene ID	NIP_FPKM	mir156abcdghijkl _FPKM	Up or down	P _value	q _value
	ZEP1	LOC_Os04g37619	2.16632	4.3149	up	0.00015	0.00357622
	NCED1	LOC_Os02g47510	3.5255	8.82215	up	0.00005	0.00139824
	MHZ4	LOC_Os01g03750	21.5725	21.248	down	0.9198	0.981002
ABA biosynthetic genes	NCED2	LOC_Os12g24800	—	—	—	—	—
	NCED3	LOC_Os03g44380	4.67463	3.06745	down	0.0142	0.11194
	NCED4	LOC_Os07g05940	—	—	—	—	—
	NCED5	LOC_Os12g42280	5.13339	4.04779	down	0.14615	0.496831
	ABA2	LOC_Os03g59610	13.63	12.3085	down	0.4784	0.825549
	DSM2	LOC_Os03g03370	8.46644	8.89035	up	0.771	0.947124
ABA deactivating genes	ABA8ox1	LOC_Os02g47470	5.34071	3.48866	down	0.00925	0.0838976
	ABA8ox2	LOC_Os08g36860	12.0595	7.9266	down	0.00355	0.0416264
	ABA8ox3	LOC_Os09g28390	—	—	—	—	—

The data were taken from the transcriptome analyses. —, undetectable expression; NIP, Nipponbare.